# Formicine ants swallow their highly acidic poison for gut microbial selection and control

**Simon Tragust[1][†]\*, Claudia Herrmann[1], Jane Häfner[1], Ronja Braasch[1], Christina Tilgen[1], Maria Hoock[1], Margarita Artemis Milidakis[1], Roy Gross[2], Heike Feldhaar[1]**

[1]Animal Ecology I, Bayreuth Center for Ecology and Environmental Research (BayCEER), University of Bayreuth, Universitätsstraße, Bayreuth, Germany; [2]Microbiology, Biocenter, University of Würzburg, Am Hubland, Würzburg, Germany

**Abstract** Animals continuously encounter microorganisms that are essential for health or cause disease. They are thus challenged to control harmful microbes while allowing the acquisition of beneficial microbes. This challenge is likely especially important for social insects with respect to microbes in food, as they often store food and exchange food among colony members. Here we show that formicine ants actively swallow their antimicrobial, highly acidic poison gland secretion. The ensuing acidic environment in the stomach, the crop, can limit the establishment of pathogenic and opportunistic microbes ingested with food and improve the survival of ants when faced with pathogen contaminated food. At the same time, crop acidity selectively allows acquisition and colonization by Acetobacteraceae, known bacterial gut associates of formicine ants. This suggests that swallowing of the poison in formicine ants acts as a microbial filter and that antimicrobials have a potentially widespread but so far underappreciated dual role in host-microbe interactions.

**\*For correspondence:**
simon.tragust@zoologie.uni-halle.de

**Present address:** [†]General Zoology, Hoher Weg 8, Martin-Luther University, Halle, Germany

**Competing interests:** The authors declare that no competing interests exist.

## Introduction

Animals commonly harbor gut-associated microbial communities (*Engel and Moran, 2013*; *Moran et al., 2019*). Patterns of recurring gut microbial communities have been described for many animal groups (*Brune and Dietrich, 2015*; *Kwong et al., 2017*; *Ochman et al., 2010*). The processes generating these patterns are however often not well understood. They might result from host filtering (*Mazel et al., 2018*), a shared evolutionary history between gut-associated microbes and their hosts (*Moeller et al., 2016*) involving microbial adaptations to the host environment (*McFall-Ngai et al., 2013*), simply be a byproduct of similar host dietary preferences (*Anderson et al., 2012*; *Hammer et al., 2017*), or result from interactions between microbes in the gut-associated microbial community (*Brinker et al., 2019*; *García-Bayona and Comstock, 2018*).

Food is an important environmental source of microbial gut associates (*Blum et al., 2013*; *Broderick and Lemaitre, 2012*; *David et al., 2014*; *Hammer et al., 2017*; *Perez-Cobas et al., 2015*; *Pais et al., 2018*) but also poses a challenge, the need to discriminate between harmful and beneficial microbes, as food may contain microbes that produce toxic chemicals or that are pathogenic (*Burkepile et al., 2006*; *Demain and Fang, 2000*; *Janzen, 1977*; *Trienens et al., 2010*). In social animals, control of harmful microbes in food while at the same time allowing the acquisition and transmission of beneficial microbes from and with food, is likely especially important. Eusocial Hymenoptera not only transport and store food in their stomach, the crop, but also distribute food to members of their colony via trophallaxis, i.e. the regurgitation of crop content from donor individuals to receiver individuals through mouth-to-mouth feeding (*Gernat et al., 2018*; *Greenwald et al.,*

*2018*; *LeBoeuf et al., 2016*). While trophallaxis can facilitate the transmission of beneficial microbes, it can also entail significant costs, as it might open the door to unwanted microbial opportunists and pathogens that can take advantage of these transmission routes (*Onchuru et al., 2018*; *Salem et al., 2015*).

Here we investigate how formicine ants, specifically the Florida carpenter ant *Camponotus floridanus*, solve the challenge to control harmful microbes in their food while allowing acquisition and transmission of beneficial microbes from and with their food. Apart from specialized intracellular endosymbionts associated with the midgut in the ant tribe Camponotini (*Degnan et al., 2004*; *Feldhaar et al., 2007*; *Russell et al., 2017*; *Williams and Wernegreen, 2015*), formicine ant species have only low abundances of microbial associates in their gut lumen but carry members of the bacterial family Acetobacteraceae as a recurring part of their gut microbiota (*Brown and Wernegreen, 2016*; *Chua et al., 2018*; *He et al., 2011*; *Ivens et al., 2018*; *Russell et al., 2017*). Some formicine gut-associated Acetobacteraceae show signs of genomic and metabolic adaptations to their host environment indicating coevolution and mutual benefit (*Brown and Wernegreen, 2019*; *Chua et al., 2020*). But the recurrent presence of Acetobacteraceae in the gut of formicine ants potentially also reflects the direct transmission of bacteria among individuals, selective uptake on the part of the ants, specific adaptations for colonizing ant guts on the part of the bacteria, or some combination of all three (*Engel and Moran, 2013*).

Formicine ant species possess a highly acidic poison gland secretion containing formic acid as its main component (*Lopez et al., 1993*; *Osman and Brander, 1961*; *Schmidt, 1986*). Although the poison is presumably foremost used as a defensive weapon (*Osman and Kloft, 1961*), it is also distributed to the environment of these ants as an external immune defense trait (sensu *Otti et al., 2014*) to protect their offspring and the nest and to limit disease spread within the society (see references in *Tragust, 2016*; *Brütsch et al., 2017*; *Pull et al., 2018*). To this end, ants take up their poison from the acidopore, the opening of the poison gland at the gaster tip, into their mouth (*Tragust et al., 2013*) during a specialized behavior existing only in a subset of ant families among all Hymenopterans (*Basibuyuk and Quicke, 1999*; *Farish, 1972*), termed acidopore grooming.

Here we first investigate whether the poison is also swallowed during acidopore grooming in *C. floridanus* and seven other formicine ant species from three genera in a comparative survey. In survival experiments and in in vitro and in vivo bacterial viability and growth experiments, we then investigate whether swallowing of the poison can serve gut microbial control and may prevent bacterial pathogen infection. Complementing these experiments, we also test whether poison swallowing has the potential to limit pathogen transmission during trophallactic food exchange. Finally, we explore whether swallowing of the poison acts as a microbial filter that is permissible to gut colonization by bacteria from the family Acetobacteraceae.

## Results

### Swallowing of the formicine ant poison gland secretion leads to acidic crop environments

To reveal whether formicine ants swallow their acidic poison during acidopore grooming, we first monitored acidity levels in the crop lumen of the Florida carpenter ant *Camponotus floridanus* after feeding them 10% honey water (pH = 5). We found that after feeding the crop became increasingly acidic over time, reaching highly acidic values 48 hr after feeding (median pH = 2; 95% CI: 1.5–3.4), whilst renewed access to food after 48 hr raised the pH to levels recorded after the first feeding (*Figure 1a*; LMM, LR-test, $\chi^2$ = 315.18, df = 3, p<0.001; Westfall corrected post-hoc comparisons: 0+4 hr versus. 48h+4 hr: p=0.317, all other comparisons: p<0.001). We also found that crop pH levels of *C. floridanus* ants were highly acidic in workers taken directly out of a satiated colony (*Figure 1—figure supplement 1*; major workers: median pH = 2, 95% CI: 2–3; minor workers: median pH = 3, CI: 2.5–3.6) and in worker cohorts that were satiated for 3 d and then starved for 24 hr before measurements (majors: median pH = 2, 95% CI: 2–3; minors: median pH = 2, CI: 2–3), suggesting that under natural conditions an acidic baseline pH in the crop lumen is maintained following perturbation thereof through ingested fluids.

To pinpoint acidopore grooming and swallowing of the poison gland secretion as the source for crop acidity and to exclude that internal, physiological mechanisms cause crop acidity, we then

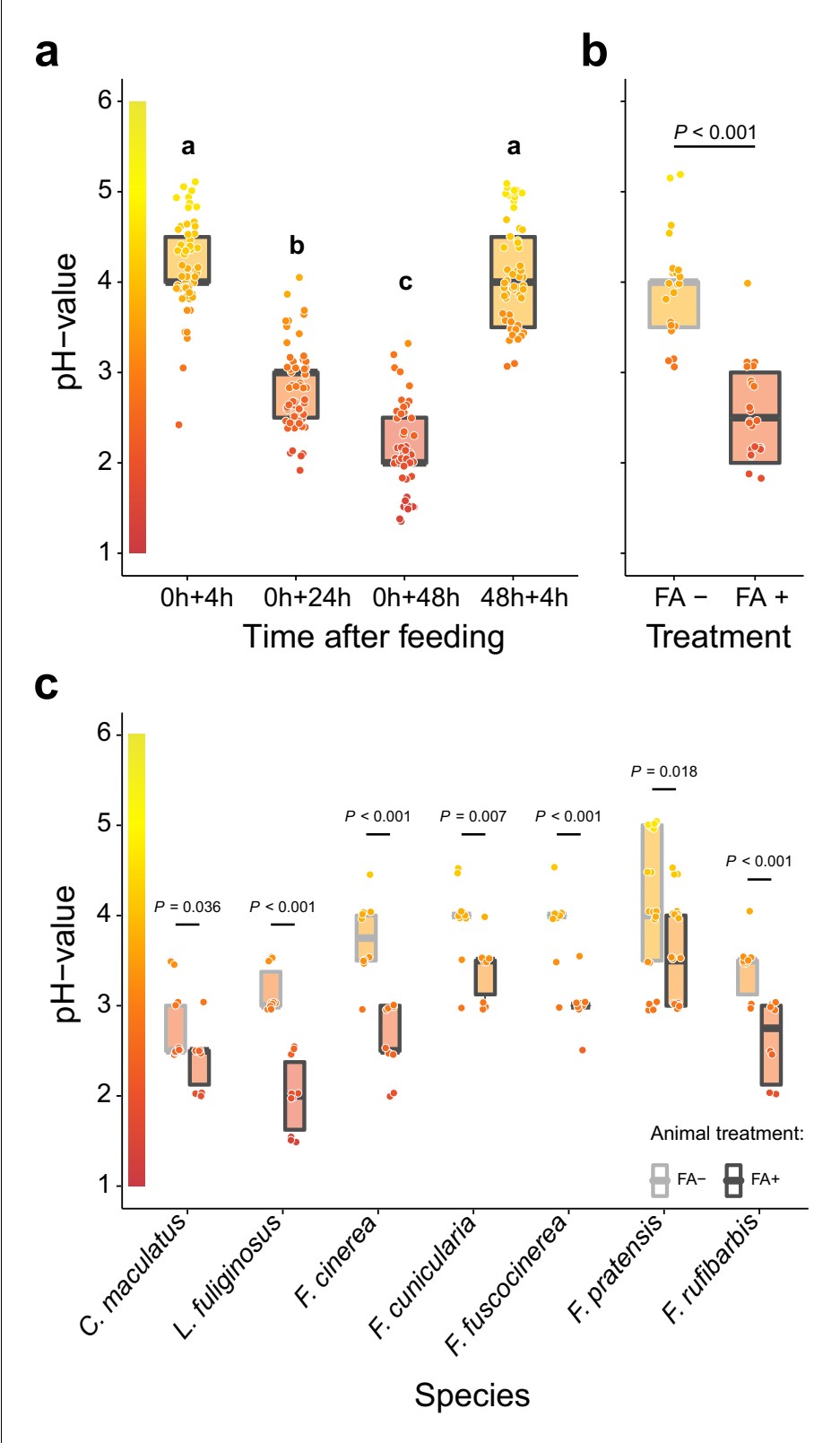

**Figure 1.** Acidification of formicine ant crop lumens through swallowing of acidic poison gland secretions. (**a**) The pH of crop lumens at 4 hr, 24 hr, and 48 hr after feeding *C. floridanus* ants 10% honey water (pH = 5) at 0 hr and at 4 hr after re-feeding ants at 48 hr (LMM, LR-test, $\chi^2$ = 315.18, df = 3, p<0.001, same letters indicate p=0.317 and different letters indicate p<0.001 in Westfall corrected post hoc comparisons). (**b**) The pH of crop lumens in *C. floridanus* ants that were either prevented to ingest the formic acid containing poison gland secretion (FA-) or not (FA+) for 24 hr after feeding

*Figure 1 continued on next page*

*Figure 1 continued*

(LMM, LR-test, $\chi^2$ = 44.68, df = 1, p<0.001). (c) The pH-value of crop lumens 24 hr after feeding in seven formicine ant species that were either prevented to ingest the formic acid containing poison gland secretion (FA-) or not (FA+). Wilcoxon rank-sum tests (two-sided). Lines and shaded boxes show the median and interquartile range; points show all data. Colors in shaded rectangles near the y-axis represent universal indicator pH colors. Color filling of shaded boxes correspond to the median pH color of x-axis groups and color filling of points correspond to universal indicator pH colors. The border of shaded boxes represents animal treatment (light gray: prevention of poison ingestion, FA-; dark gray: poison ingestion not prevented, FA+).

The online version of this article includes the following source data and figure supplement(s) for figure 1:

**Source data 1.** Source data for panel a, on pH of crop lumens at 4 hr, 24 hr, and 48 hr after feeding *C. floridanus* ants 10% honey water at 0 hr and at 4 hr after re-feeding ants at 48 hr.
**Source data 2.** Source data for panel b, on pH of crop lumens in *C. floridanus* ants that were either prevented to ingest formic acid containing poison gland secretions (FA-) or not (FA+) for 24 hr after feeding.
**Source data 3.** Source data for panel c, on pH of crop lumens 24 hr after feeding in seven formicine ant species that were either prevented to ingest formic acid containing poison gland secretions (FA-) or not (FA+).
**Figure supplement 1.** Baseline acidity of *C. floridanus* crop lumens under satiated and starved conditions.
**Figure supplement 1—source data 1.** Source Data on the baseline acidity of *C. floridanus* (major and minor worker caste) under satiated and starved conditions.
**Figure supplement 2.** Acidopore grooming frequency of *C. floridanus* after ingestion of different food types.
**Figure supplement 2—source data 1.** Source data on the frequency of acidopore grooming in *C. floridanus* ants within 30 min.
**Figure supplement 3.** Acidity along the gastrointestinal tract of *C. floridanus*.
**Figure supplement 3—source data 1.** Source data on pH-measurements 24 hr after access to 10% honey-water in the crop and directly after the proventriculus at four points along the midgut of *C. floridanus* ants.

prevented acidopore grooming in *C. floridanus* ants for 24 hr after feeding through immobilization. This experiment revealed that acidopore grooming prevented ants showed a significantly diminished acidity in their crop compared to ants that were not prevented from acidopore grooming (*Figure 1b*; LMM, LR-test, $\chi^2$ = 44.68, df = 1, p<0.001). A similar, significantly diminished acidity in crop lumens was ubiquitously obtained in a comparative survey across seven formicine ant species and three genera (*Camponotus*, *Lasius*, and *Formica*) upon comparison of ants that were prevented from acidopore grooming through immobilization to non-prevented ants (*Figure 1c*; two-sided Wilcoxon rank-sum tests, comparisons for all ant species: p≤0.036). We conclude that formicine ants attain a highly acidic baseline pH in their crop lumen by taking up their poison into their mouth during acidopore grooming (*Tragust et al., 2013*), and subsequently swallowing it. The comparative survey also shows that this behavior is widespread among formicine ants.

Although venomous animals often bear a cost of venom production and express behavioural adaptations to limit venom expenditure (*Casewell et al., 2013*), we also found that *C. floridanus* ants increase the frequency of acidopore grooming within the first 30 min after ingesting fluids compared to unfed ants irrespective of the fluid's nutritional value (*Figure 1—figure supplement 2*; GLMM, LR-test, $\chi^2$ = 33.526, df = 2, p<0.001; Westfall corrected post-hoc pairwise comparisons, water versus. 10% honey-water: p=0.634, unfed versus water and unfed versus 10% honey-water: both p<0.001). Moreover, we found that the strong acidity was limited to the crop of *C. floridanus* ants and did not extend to the midgut, the primary site of digestion in insects (*Holtof et al., 2019*; *Terra and Ferreira, 1994*; *Figure 1—figure supplement 3*; pH-measurements at four points along the midgut 24 hr after access to 10% honey-water; mean ± se; midgut position 1 = 5.08 ± 0.18, midgut position 2 = 5.28 ± 0.17, midgut position 3 = 5.43 ± 0.16, midgut position 4 = 5.31 ± 0.19). Together, these results led us to hypothesize that poison acidified crop lumens in formicine ants do not primarily serve a digestive function but may serve microbial control, limiting infection by oral pathogens.

## Poison acidified crops can prevent the passage of pathogenic and opportunistic bacteria to the midgut

To investigate a potential microbial control function, we next tested whether poison acidified crop lumens can inhibit *Serratia marcescens*, an insect pathogenic bacterium (*Grimont and Grimont, 2006*), when ingested together with food and prevent its passage from the crop to the midgut in *C. floridanus* ants. To this end, we first estimated food passage times through the gut of *C. floridanus*

with fluorescent particles contained in food, as we surmised that ingested fluids need to remain in the crop for a minimum time before being passed to the midgut in order for poison swallowing and the ensuing crop acidity to take effect after perturbation of the crop pH through ingested fluids. In agreement with food passage times through the gastrointestinal tract of other ants (*Cannon, 1998*; *Kloft, 1960b*; *Kloft, 1960a*; *Howard and Tschinkel, 1981*; *Markin, 1970*), we found that only a small amount of ingested food is passed from the crop to the midgut 2–4 hr after feeding, while thereafter food is steadily passed from the crop to the midgut until 18 hr after feeding (*Figure 2— figure supplement 1*).

We then measured the viability of *Serratia marcescens* ingested together with food in the gastrointestinal tract of *C. floridanus* at two time points before (0.5 hr and 4 hr) and after (24 hr and 48 hr) main food passage from the crop to the midgut, with the time directly after food ingestion (0 hr) serving as a reference. We found that *S. marcescens* presence decreased sharply over time in the crop (*Figure 2a*; GLMM, LR-test, $\chi^2 = 220.78$, df = 4, p<0.001). The proportion of CFUs that we were able to retrieve from the crop relative to the mean at 0 hr in the crop diminished from 43% at 0.5 hr post-feeding (median, CI: 0–543%) to 0% at 4 hr (CI: 0–4%), 24 hr (CI: 0–1.8%), and 48 hr (CI: 0–18%) post-feeding. In addition, relative to the mean at 0 hr in the crop, *S. marcescens* could only be detected at extremely low numbers in the midgut (median 0%) at 0 hr (CI: 0–4%), 0.5 hr (CI: 0– 1%) and 24 hr (CI: 0–1%) post-feeding and not at all at 4 hr and 48 hr post-feeding (*Figure 2b*; GLMM, LR-test, $\chi^2 = 1.044$, df = 2, p=0.593). A similar, rapid reduction in the crop and inability to pass from the crop to the midgut was obtained when we fed *E. coli*, a potential opportunistic bacterium that is not a gut associate of insects (*Blount, 2015*) to *C. floridanus* ants (*Figure 2—figure supplement 2*; crop: GLMM, LR-test, $\chi^2 = 156.74$, df = 4, p<0.001; midgut: GLMM, LR-test, $\chi^2 = 14.898$, df = 3, p=0.002).

Although in vivo the antimicrobial activity of the natural poison is likely higher than the antimicrobial activity of formic acid, the main component of the formicine poison gland secretion (*Lopez et al., 1993*; *Osman and Brander, 1961*; *Schmidt, 1986*) due to the presence of other components (*Tragust et al., 2013*), we then tested the ability of *S. marcescens* to withstand acidic conditions created with formic acid in an in vitro experiment. We found that incubation of *S. marcescens* for 2 hr in 10% honey water acidified with formic acid to pH 4 resulted in a significantly lower number of CFUs relative to pH 5 and in zero growth for incubations at pH-levels that were lower than 4 (*Figure 2—figure supplement 3*; GLM, LR-test, $\chi^2 = 79.442$, df = 1, p<0.001). Our data thus indicate that poison acidified crops can indeed serve microbial control in formicine ants, likely inhibiting bacteria according to their ability to cope with acidic environments (*Lund et al., 2014*).

## Access to the poison improves survival upon ingestion of pathogen contaminated food

To test whether acidic crops also provide a fitness benefit upon ingestion of pathogen contaminated food, we prevented acidopore grooming through immobilization in *C. floridanus* ants for 24 hr after feeding them once with 5 µL of either *S. marcescens* contaminated honey water or non-contaminated honey water and monitored their survival thereafter without providing additional food. We found that acidopore access after pathogen ingestion significantly increased the survival probability of ants (*Figure 3*; COXME, LR-test, $\chi^2 = 20.95$, df = 3, p=0.0001). The survival of ants prevented from acidopore grooming and fed once with the pathogen contaminated food was significantly lower than that of non-prevented ants fed the same food source (Westfall corrected post-hoc comparisons: FA - | *Serratia* presence + versus. all other ant groups: p≤0.027). In contrast, non-prevented ants fed once with the pathogen contaminated food source did not differ in survival to prevented and non-prevented ants fed the non-contaminated food source (Westfall corrected post-hoc comparisons: FA + | *Serratia* presence + versus. FA + | *Serratia* presence – and FA + | *Serratia* presence + versus. FA + | *Serratia* presence –: p≥0.061 for both comparisons). Although we observed an overall high mortality in this experimental setup, likely due to starvation following the one time feeding in combination with social isolation of individually kept ants (*Kohlmeier et al., 2016*; *Koto et al., 2015*; *Stucki et al., 2019*), this result indicates that poison acidified crop lumens provide a fitness benefit in terms of survival to formicine ants upon ingestion of pathogen contaminated food.

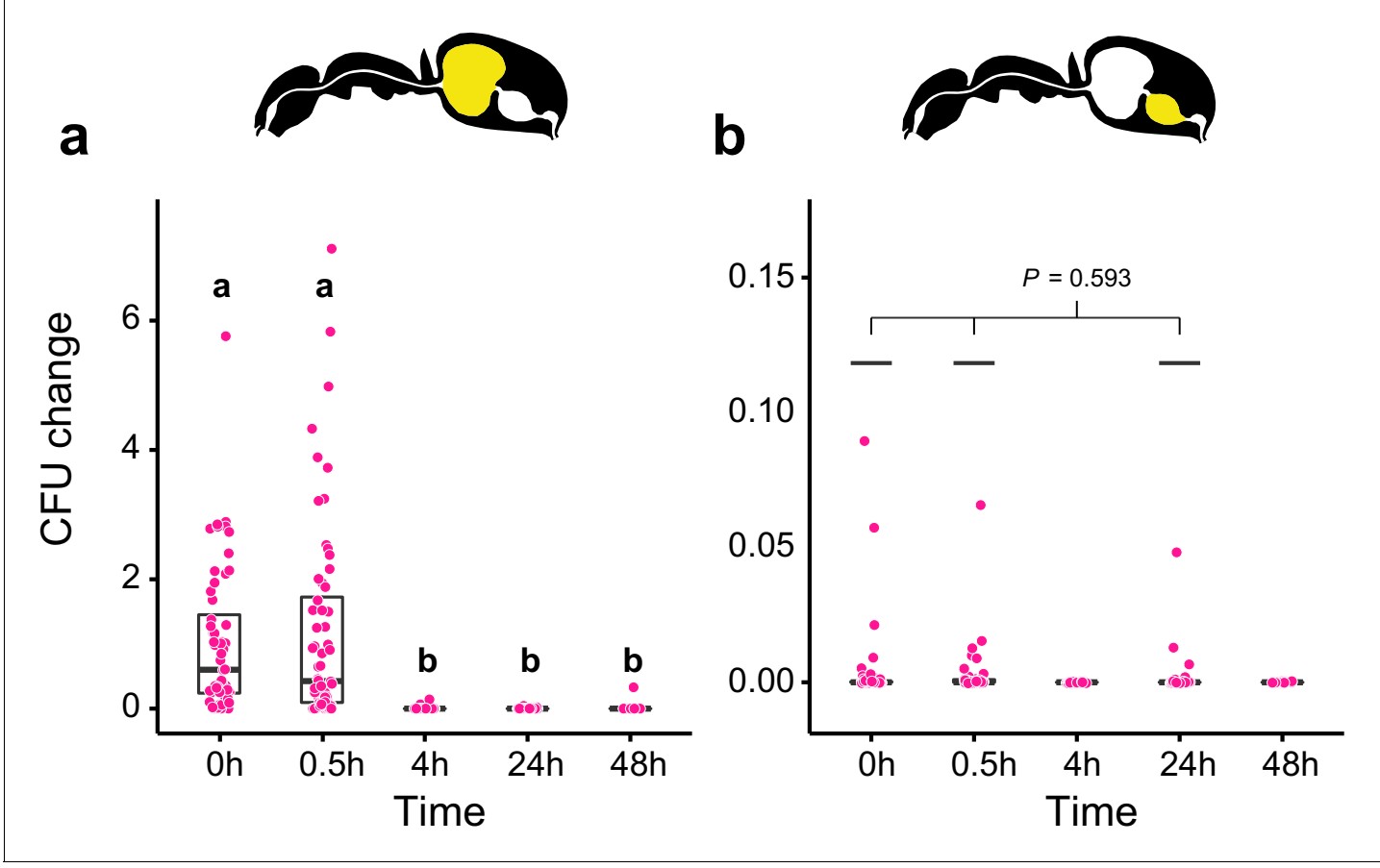

**Figure 2.** Viability of *S. marcescens* over time in the digestive tract of *C. floridanus*. Change in the number of colony forming units (CFUs) in the crop (**a**) and midgut (**b**) part of the digestive tract (yellow color in insert) relative to the mean CFU-number at 0 hr in the crop (CFU change corresponds to single data CFU-value divided by mean CFU-value at 0 hr in the crop), 0 hr, 0.5 hr, 4 hr, 24 hr, and 48 hr after feeding *Camponotus floridanus* ants 10% honey water contaminated with *Serratia marcescens*. (**a**), Change of *S. marcescens* in the crop (GLMM, LR-test, $\chi^2$ = 220.78, df = 4, p<0.001, same letters indicate p≥0.623 and different letters indicate p<0.001 in Westfall corrected post hoc comparisons). (**b**), Change of *S. marcescens* in the midgut (GLMM, LR-test, $\chi^2$ = 1.044, df = 2, p=0.593). Note that timepoints with zero bacterial growth in the midgut (4 hr and 48 hr) were excluded from the statistical model.

The online version of this article includes the following source data and figure supplement(s) for figure 2:

**Source data 1.** Source data for panels a and b, on the number and the change in the number of colony forming units (CFUs) relative to 0 hr in the crop in the crop.

**Figure supplement 1.** Food passage of fluorescent particles through the digestive tract of *C. floridanus*.

**Figure supplement 1—source data 1.** Source data for panels a and b, on the food passage of florescent particles through the digestive tract (crop, midgut, hindgut) of *C. floridanus* minor (**a**) and major (**b**) worker ants.

**Figure supplement 2.** Viability of *E. coli* over time in the digestive tract of *C. floridanus* over time.

**Figure supplement 2—source data 1.** Source data for panels a and b, on the number and the change in the number of colony forming units (CFUs) in the crop (**a**) and midgut (**b**) part of the digestive tract of *C. floridanus* ants relative to 0 hr in the crop at 0 hr, 0.5 hr, 4 hr, 24 hr, and 48 hr after feeding ants 10% honey water contaminated with *Escherichia coli*.

**Figure supplement 3.** *S. marcescens* growth in vitro.

**Figure supplement 3—source data 1.** Source data on the number and the change in the number of CFUs relative to pH 5 after incubation of *Serratia marcescens* in 10% honey water (pH = 5) or in 10% honey water acidified with commercial formic acid to a pH of 4, 3, or 2 for 2 hr.

## Access to the poison in donor ants also benefits receiver ants without poison access after food exchange via trophallaxis

The ability to swallow the acidic poison may not only improve survival of formicine ants feeding directly on pathogen contaminated food but also of ants that share the contaminated food via trophallaxis. To test this, we created two types of donor-receiver ant pairs. Donor ants in both pairs

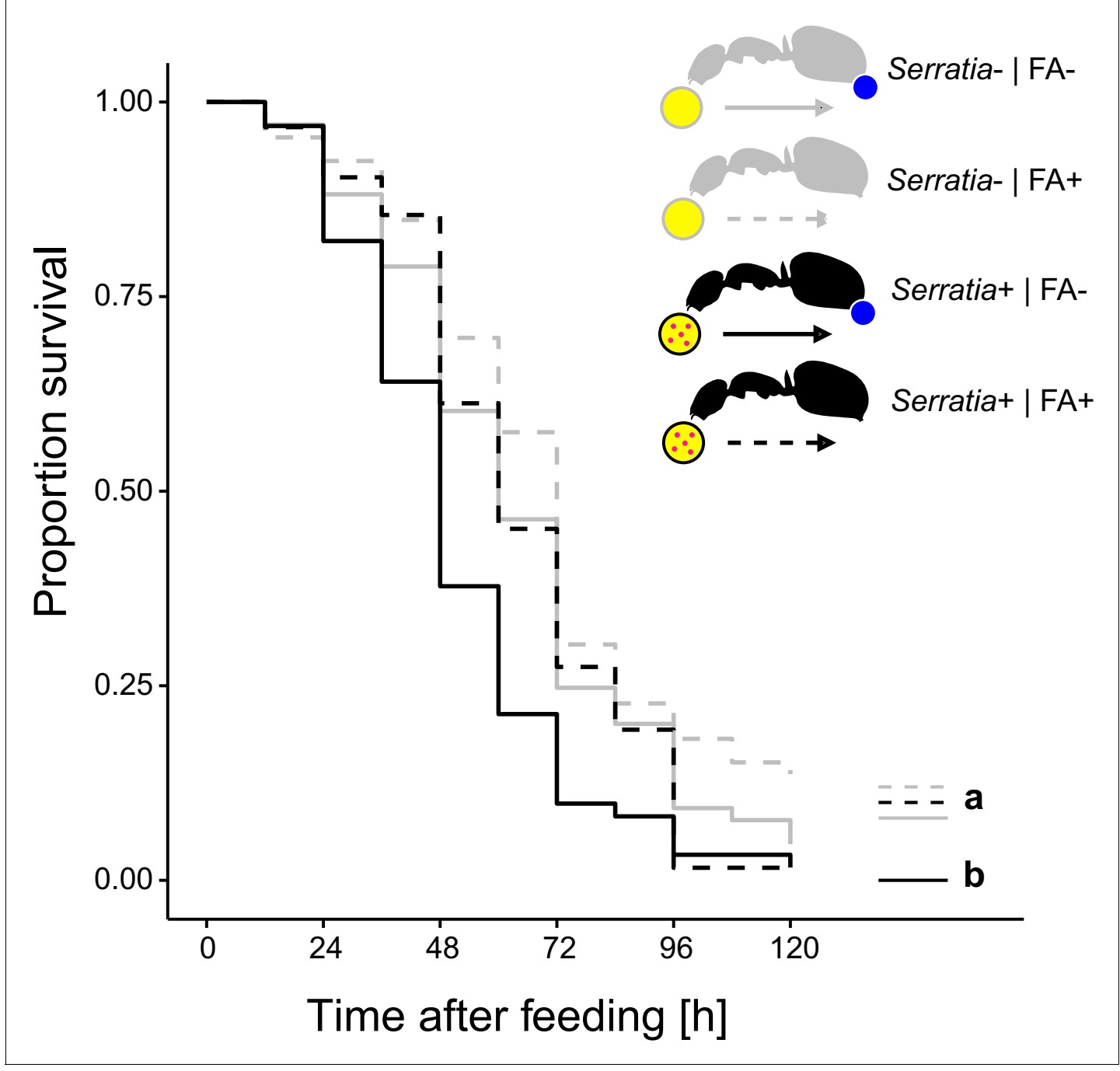

**Figure 3.** Survival after ingestion of pathogen contaminated food. Survival of individual *C. floridanus* ants that were either prevented to ingest the formic acid containing poison gland secretion (FA-; ant outlines with blue dot) or not (FA+) after feeding them once either honey water contaminated with *Serratia marcescens* (*Serratia*+, yellow circle with pink dots and black ant outlines) or non-contaminated honey water (*Serratia*-) without providing food thereafter (COXME, LR-test, $\chi^2$ = 20.95, df = 3, p=0.0001, same letters indicate p≥0.061 and different letters indicate p≤0.027 in Westfall corrected post hoc comparisons).

The online version of this article includes the following source data for figure 3:

**Source data 1.** Source data on the survival of individual *C. floridanus* ants that were either prevented to ingest formic acid containing poison gland secretions (FA-) or not (FA+) after feeding on either honey water contaminated with *Serratia marcescens* (*Serratia*+) or non-contaminated honey water (*Serratia*-).

were directly fed *S. marcescens* contaminated food every other day, while receiver ants obtained food only through trophallaxis from their respective donor ants. Receiver ants in both pairs were precluded from swallowing of the poison through blockage of their acidopore opening, while donor ants were blocked in one pair but only sham blocked in the other pair. We found that the duration of trophallaxis between the two donor-receiver ant pairs during the first 30 min. of the first feeding bout did not significantly differ (*Figure 4—figure supplement 1*; LMM, LR-test, $\chi^2$ = 1.23, df = 1, p=0.268), indicating that trophallactic behavior was not influenced through acidopore blockage in donor ants at the beginning of the experiment. Over the next 12 d, we found that acidopore blockage per se had a significant negative effect on the survival of donor as well as receiver ants (*Figure 4*; COXME, LR-test, $\chi^2$ = 66.68, df = 3, p<0.001). However, although receiver ants that obtained food every other day from donors with the ability to swallow the poison died at a higher rate than their respective donor counterparts (hazard ratio: 1.81; Westfall corrected post-hoc comparison: p<0.001) they were only half as likely to die compared to receiver ants that obtained pathogen contaminated food from blocked donors unable to swallow their poison (hazard ratio: 0.56; Westfall corrected post-hoc comparison: p<0.001). This indicates that swallowing of the poison and the ensuing crop acidity also provides a fitness benefit to other members of a formicine ant society.

## Poison acidified crops allow members of the bacteria family Acetobacteraceae passage to the midgut

In addition to microbial control, poison acidified formicine ant crops might act as a chemical filter for gut-associated microbial communities, similar to gut morphological structures that can act as mechanical filters in ants (*Cannon, 1998*; *Glancey et al., 1981*; *Lanan et al., 2016*; *Little et al., 2006*; *Quinlan and Cherrett, 1978*) and other insects (*Itoh et al., 2019*; *Ohbayashi et al., 2015*). To investigate the idea of a chemical filter, we tested the ability of the insect gut-associated bacterium *Asaia* sp. (family Acetobacteraceae) (*Crotti et al., 2009*; *Favia et al., 2007*) to withstand acidic environments in vitro and in vivo. In contrast to *S. marcescens* (*Figure 2—figure supplement 2*), *Asaia* sp. was not affected by an incubation for 2 hr in 10% honey water acidified with formic acid to pH 4 and was still able to grow when incubated at pH 3 in in vitro tests (*Figure 5—figure supplement 1*; GLM, overall LR-test $\chi^2$ = 21.179, df = 2, p<0.001; Westfall corrected post hoc comparisons: pH = 5 versus. pH = 4: p=0.234, all other comparisons: p<0.001). Moreover, in in vivo tests, *Asaia* sp. only gradually diminished over time in the crop (*Figure 5a*; GLMM; LR-test, $\chi^2$ = 124.01, df = 4, p<0.001) with the proportion of CFUs that we were able to retrieve from the crop relative to the mean at 0 hr in the crop diminishing to only 34% (median, CI: 3–85%) and 2% (CI: 0–7%) at 4 hr and 24 hr post-feeding, respectively. At the same time, relative to the mean at 0 hr in the crop, *Asaia* sp. steadily increased in the midgut (*Figure 5b*; GLMM; LR-test, $\chi^2$ = 59.94, df = 3, p<0.001) from its initial absence at 0 hr post-feeding to 2% (median, CI: 0–5%) at 48 hr post-feeding. This suggests that in formicine ants, poison acidified crops might act as a chemical filter that works selectively against the establishment of opportunistic and potentially harmful bacteria but allows entry and establishment of members of the bacterial family Acetobacteraceae.

## Discussion

In this study, we investigated how formicine ants solve the challenge to control harmful microbes in their food while at the same time allowing acquisition and transmission of beneficial microbes from and with their food. We found that formicine ants swallow their antimicrobial, highly acidic poison gland secretion during the behavior of acidopore grooming. The resulting acidic environment in their stomach, the crop, can protect formicine ants from food borne bacterial pathogens while at the same time allowing the acquisition and establishment of members of the bacterial family Acetobacteraceae, a recurring part of the gut microbiota of formicine ants.

Highly acidic stomach lumens are ubiquitous in higher vertebrates, including amphibians, reptiles, birds and mammals (*Beasley et al., 2015*; *Koelz, 1992*). In insects, highly acidic gut regions have so far only rarely been described from the midgut (*Chapman, 2013*; *Holtof et al., 2019*). The mechanisms responsible for the creation of a gut lumen compartment with a certain pH are often unknown in insects (*Harrison, 2001*), but in principle, highly acidic gut regions in insects may, similar to vertebrates (*Hersey and Sachs, 1995*), be generated through physiological mechanisms (*Matthews, 2017*; *Miguel-Aliaga et al., 2018*; *Onken and Moffett, 2017*). Alternatively, acidic

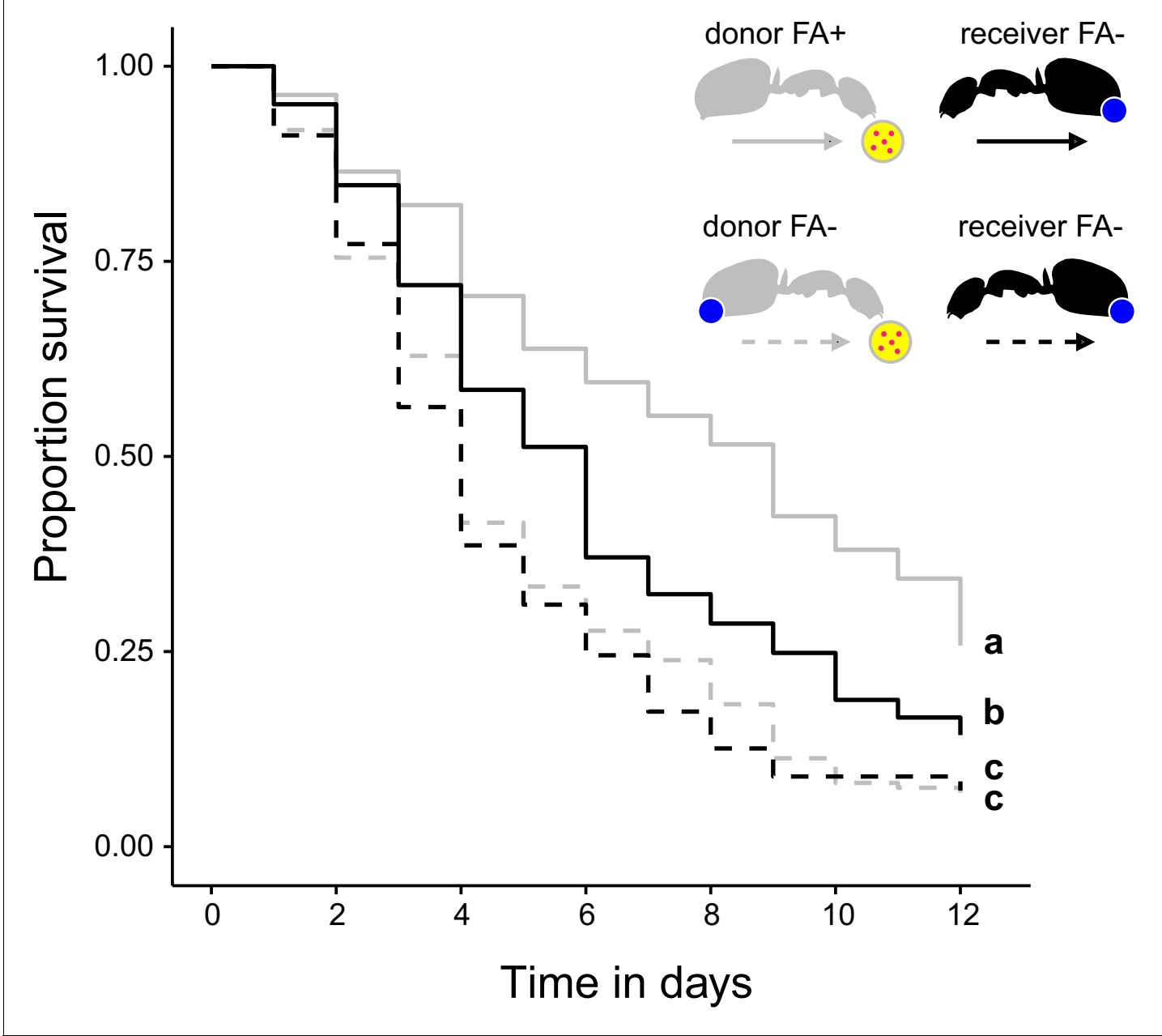

**Figure 4.** Survival after sharing pathogen contaminated food via trophallaxis. Survival of donor ants (light gray ant outlines) that were directly fed with pathogen contaminated food (yellow circle with pink dots in insert) every other day and were either prevented to ingest their formic acid containing poison gland secretion (FA-; ant outlines with blue dot) or not (FA+) and survival of receiver ants (black ant outlines) that received pathogen contaminated food only through trophallaxis with donor ants and were always prevented to ingest their formic acid containing poison gland secretion (FA-) (COXME, LR-test, $\chi^2$ = 66.68, df = 3, p<0.001, same letters indicate p=0.309 and different letters indicate p≤0.002 in Westfall corrected post hoc comparisons).

The online version of this article includes the following source data and figure supplement(s) for figure 4:

**Source data 1.** Source data on the survival of donor *C. floridanus* ants that were directly fed with pathogen contaminated food and were either prevented to ingest formic acid containing poison gland secretions (FA-) or not (FA+) and survival of receiver ants that received pathogen contaminated food only through trophallaxis with donor ants and were always prevented to ingest formic acid containing poison gland secretions (FA-).

**Figure supplement 1.** Duration of trophallaxis in donor-receiver ant pairs.

**Figure supplement 1—source data 1.** Total duration of trophallaxis events within 30 min.

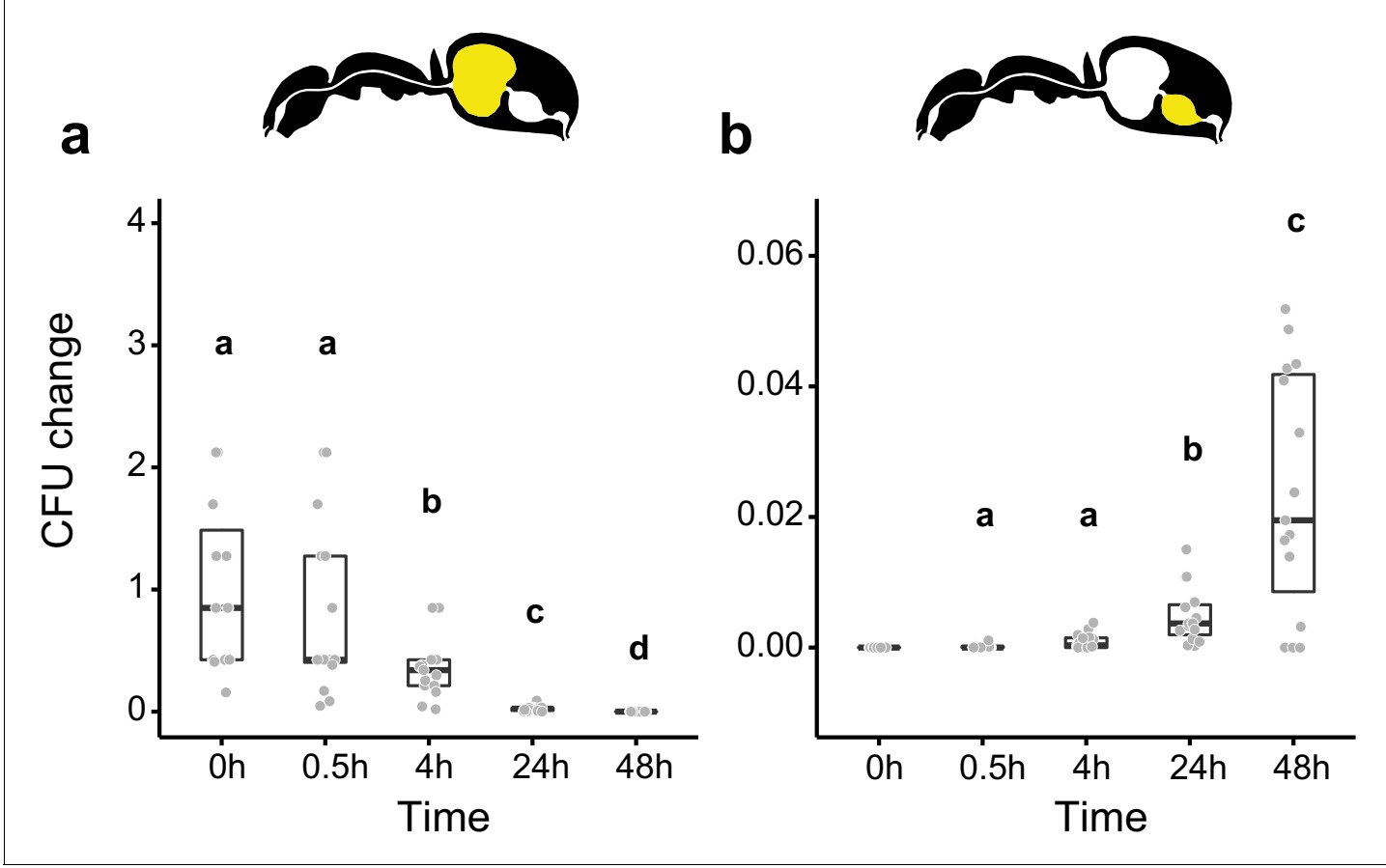

**Figure 5.** Viability of *Asaia sp.* over time in the digestive tract of *C. floridanus*. Change in the number of colony forming units (CFUs) in the crop (**a**) and midgut (**b**) part of the digestive tract (yellow color in insert) relative to the mean CFU-number at 0 hr in the crop (CFU change corresponds to single data CFU-values divided by mean CFU-value at 0 hr in the crop), 0 hr, 0.5 hr, 4 hr, 24 hr, and 48 hr after feeding ants 10% honey water contaminated with *Asaia* sp. (**a**), Change of *Asaia* sp. in the crop (GLMM; LR-test, $\chi^2$ = 124.01, df = 4, p<0.001, same letters indicate p=0.488 and different letters indicate p≤0.013 in Westfall corrected post hoc comparisons). (**b**), Change of *Asaia* sp. in the midgut (GLMM; LR-test, $\chi^2$ = 59.94, df = 3, p<0.001, same letters indicate p=0.116 and different letters indicate p≤0.005 in Westfall corrected post hoc comparisons). Note that timepoints with zero bacterial growth in the midgut (0 hr) were excluded from the statistical model.

The online version of this article includes the following source data and figure supplement(s) for figure 5:

**Source data 1.** Source data for panels a and b, on the number and the change in the number of colony forming units (CFUs) relative to 0 hr in the crop in the crop (**a**) and midgut (**b**) part of the digestive tract of *C. floridanus* ants at 0 hr, 0.5 hr, 4 hr, 24 hr, and 48 hr after feeding ants 10% honey water contaminated with *Asaia* sp.

**Figure supplement 1.** *Asaia sp.* growth in vitro.

**Figure supplement 1—source data 1.** Source data on the number and the change in the number of CFUs relative to pH five after incubation of *Asaia* sp.

derivatives of gut-associated microbes (*Ratzke et al., 2018*, *Ratzke and Gore, 2018*, *Wolfe, 2005*) or acidic gland secretions (*Blum, 1996*; *Morgan, 2008*; *Vander Meer, 2012*) might contribute to the insect gut pH. In agreement with the latter, the results of our study show that formicine ants maintain a highly acidic baseline pH in their stomach, the crop, through swallowing of their poison gland secretion during acidopore grooming. Interestingly, although we found that a higher crop acidity was observed in all formicine ants in our comparative survey when they had access to their poison, we also found that crop acidity was highly variable in ants with and without access to their poison. While a variable crop acidity in ants without access to their poison could indicate the existence of additional internal or external sources that maintain crop acidity, a variable crop acidity in ants with access to their poison could indicate species specific differences in acidopore grooming, in

the composition of the poison gland secretion or in optimal crop acidity. Future studies will need to explore these possibilities.

Sanitation of food through the addition of organic acids or through acidic fermentation is frequently practiced by humans (*Cherrington et al., 1991*; *Hirshfield et al., 2003*; *Theron and Rykers Lues, 2010*) and sanitation of food with antimicrobials from different sources is ubiquitous in animals that provision food to their offspring or that store, cultivate, develop or live in food (*Currie et al., 1999*; *Herzner et al., 2013*; *Herzner and Strohm, 2007*; *Joop et al., 2014*; *Mueller et al., 2005*; *Vander Wall, 1990*). A microbial control function of poison acidified crops in formicine ants to sanitize ingested food is supported by our survival experiments and our in vivo and in vitro bacterial growth and viability experiments. There we found that access to the poison improved survival of formicine ants after feeding on pathogen contaminated food. We also found that pathogenic and opportunistic bacteria were quickly inhibited in the crop when ingested with food and could not establish in the midgut. Although our data suggests that this is likely due to the sensitivity of these bacteria to acidic environments, our evidence for this is only indirect. At present it is unclear whether the acidic environment in the crop is sufficient to protect formicine ants and to inhibit pathogenic and opportunistic microbes ingested with food or whether acidic conditions act in concert with other factors. Studies in vertebrates and the fruit fly *Drosophila melanogaster* have shown that acidic gut regions together with immune system effectors serve microbial control and prevent infection by oral pathogens (*Giannella et al., 1972*; *Howden and Hunt, 1987*; *Martinsen et al., 2005*; *Overend et al., 2016*; *Rakoff-Nahoum et al., 2004*; *Slack et al., 2009*; *Tennant et al., 2008*; *Watnick and Jugder, 2020*). Concordantly, previous studies investigating formicine ant trophallactic fluids *Hamilton et al., 2011*; *LeBoeuf et al., 2016* found the presence of proteins related to cathepsin D, a lysosomal aspartic protease that can exhibit antibacterial effector activity and the proteolytic production of antimicrobial peptides (*Ning et al., 2018*). Future studies will therefore need to disentangle the relative contributions of crop acidity and immune system effectors released into the gut lumen to the improved survival of formicine ants in the face of pathogen contaminated food and to the microbe inhibitory action of poison acidified crops in formicine ants.

In addition to improving their own survival, the ability of donor ants to access their poison also improved the survival of receiver ants without access to their poison following trophallactic exchange of pathogen-contaminated food. Acidic crop lumens might therefore act as a barrier to disease spread in formicine ant societies, alleviating the cost of sharing pathogen contaminated food (*Onchuru et al., 2018*; *Salem et al., 2015*) and counteracting the generally increased risk of pathogen exposure and transmission associated with group-living (*Alexander, 1974*; *Boomsma et al., 2005*; *Kappeler et al., 2015*). Although food distribution via trophallaxis is a dynamic process governed by many different factors (*Buffin et al., 2009*; *Buffin et al., 2011*; *Greenwald et al., 2015*; *Sendova-Franks et al., 2010*), the technological advances in recent years to track multiple individuals of a group simultaneously over time (*Gernat et al., 2018*; *Greenwald et al., 2015*; *Imirzian et al., 2019*; *Stroeymeyt et al., 2018*), will make it possible to clarify the contribution of acidic crop lumens to disease spread prevention in formicine ant societies.

Acidic crop lumens might not only serve microbial control but might also act as a chemical filter for microbes, working selectively against pathogenic or opportunistic bacteria but allowing entry and establishment of species from the bacteria family Acetobacteraceae. We found that, compared to a bacterial pathogen, a bacterial member of the Acetobacteraceae was not only better able to withstand acidic conditions created with formic acid in vitro but was able to establish itself in the midgut of formicine ants in vivo. This suggests that host filtering of microbes (*Mazel et al., 2018*) via acidic crop lumens might explain at least part of the recurrent presence of Acetobacteraceae in the gut of formicine ants and the otherwise reduced microbial diversity and abundance of gut-associated microbes in formicine ants (*Brown and Wernegreen, 2016*; *Brown and Wernegreen, 2019*; *Chua et al., 2018*; *Chua et al., 2020*; *Ivens et al., 2018*; *Russell et al., 2017*).

Though not formally established (see *Mushegian and Ebert, 2016*), recent studies indicate a mutualistic relationship between formicine ants, and their gut-associated Acetobacteraceae (*Brown and Wernegreen, 2019*; *Chua et al., 2020*). Thus, the creation of an acidic crop environment in formicine ants that is easier to endure if colonizing microbes are mutualists agrees with the concept of screening as a mechanism to choose microbial partners out of a pool of environmental microbes (*Archetti et al., 2011a*; *Archetti et al., 2011b*; *Biedermann and Kaltenpoth, 2014*; *Scheuring and Yu, 2012*). Contrary to signalling, where costly information is displayed to partners,

in screening a costly environment is imposed on partners that excludes all but high-quality ones. Partner choice in a number of cross-kingdom mutualisms is readily explained by screening (see examples in *Archetti et al., 2011a*; *Archetti et al., 2011b*; *Biedermann and Kaltenpoth, 2014*; *Scheuring and Yu, 2012*) but experimental evidence is so far limited in insect-microbe associations (*Innocent et al., 2018*; *Itoh et al., 2019*; *Ranger et al., 2018*). Although our experiments can only hint at screening as a means of partner choice in formicine ants, the results of our study would provide support for the prediction that screening is more likely to evolve from a host's defense trait against parasites (*Archetti et al., 2011a*; *Archetti et al., 2011b*), that is, the highly acidic, antimicrobial poison that creates a selective environment for microbes. Our study might therefore not only provide evidence that the well-established cross talk between the immune system and gut-associated microbes in vertebrates and invertebrates (*Chu and Mazmanian, 2013*; *Rakoff-Nahoum et al., 2004*; *Slack et al., 2009*; *Watnick and Jugder, 2020*; *Xiao et al., 2019*) can hold for a broader range of immune defense traits (sensu *Otti et al., 2014*) but also that this cross talk can be realized through signals (*Fischbach and Segre, 2016*; *Moura-Alves et al., 2019*; *Villena et al., 2018*) and through screening.

Overall, our study provides evidence that poison acidified crop lumens of formicine ants can act as a chemical filter for control and selection of microbes ingested with food. Poison acidified formicine crops might thus contribute to the ecological and evolutionary success of this group of insects by alleviating the increased risk of pathogen exposure and transmission associated with group living but allowing the acquisition and transmission of microbial mutualists. Similar microbial filters likely represent a widespread theme to manage harmful and beneficial host-associated microbes but have so far only partly been uncovered in a few animal systems (*Cardoza et al., 2006*; *Duarte et al., 2018*; *Sapountzis et al., 2019*; *Scott et al., 2008*; *Shukla et al., 2018a*; *Shukla et al., 2018b*; *Vogel et al., 2017*).

# Materials and methods

**Key resources table**

| Reagent type (species) or resource | Designation | Source or reference | Identifiers | Additional information |
|---|---|---|---|---|
| Biological sample (*Camponotus floridanus*) | *Camponotus floridanus* | other | | See Materials and methods |
| Biological sample (*Camponotus maculatus*) | *Camponotus maculatus* | other | | See Materials and methods |
| Biological sample (*Lasius fuliginosus*) | *Lasius fuliginosus* | other | | See Materials and methods |
| Biological sample (*Formica cinerea*) | *Formica cinerea* | other | | See Materials and methods |
| Biological sample (*Formica cunicularia*) | Formica cunicularia | other | | See Materials and methods |
| Biological sample (*Formica fuscocinerea*) | Formica fuscocinerea | other | | See Materials and methods |
| Biological sample (*Formica pratensis*) | Formica pratensis | other | | See Materials and methods |
| Biological sample (*Formica rufibarbis*) | Formica rufibarbis | other | | See Materials and methods |
| Strain, strain background (*Serratia marcescens*) | *Serratia marcescens* | Strain DSM12481, DSMZ-German Collection of Microorganisms and Cell Cultures GmbH, Braunschweig, Germany | | |

*Continued on next page*

*Continued*

| Reagent type (species) or resource | Designation | Source or reference | Identifiers | Additional information |
|---|---|---|---|---|
| Strain, strain background (*Escherichia coli*) | *Escherichia coli* | Strain DSM6897, DSMZ-German Collection of Microorganisms and Cell Cultures GmbH, Braunschweig, Germany | | |
| Strain, strain background (*Asaia* sp.) | *Asaia* sp. | Strain SF2.1 **Favia et al., 2007** | | |
| Other | Blaubrand intraMARK micro pipettes | Brand, Wertheim, Germany | 708707 | |
| Other | pH sensitive paper | Hartenstein, Würzburg, Germany | PHIP | |
| Other | pH electrode | Unisense, Aarhus, Denmark | | |
| Other | Polymethylmethacrylate | | | University of Bayreuth, Animal Ecology I, group microplastic |
| Other | Leica microscope DM 2000 LED | Leica, Wetzlar, Germany | | |
| Other | Leica stereomicroscope M 165 C | Leica, Wetzlar, Germany | | |
| Other | Commercial honey | Different brands | | 10% (w/v), 1:1 honey:water |
| Other | superglue | UHU brand | | |
| Chemical compound, drug | ≥95% Formic acid | Sigmaaldrich, Merck, Darmstadt, Germany | Cat# F0507 | |
| Chemical compound, drug | Trypton | Sigmaaldrich, Merck, Darmstadt, Germany | Cat# T7293-250G | |
| Chemical compound, drug | Yeast extract | Millipore, Merck, Darmstadt, Germany | Cat# Y1625-250G | |
| Software, algorithm | R version 3. 6.1 | | | **R Development Core Team, 2019** |

## Ant species and maintenance

Colonies of the carpenter ant *Camponotus floridanus* were collected in 2001 and 2003 in Florida, USA, housed in Fluon (Whitford GmbH, Diez, Germany) coated plastic containers with plaster ground and maintained at a constant temperature of 25°C with 70% humidity and a 12 hr/12 hr light/dark cycle. They were given water ad libitum and were fed two times per week with honey water (1:1 tap water and commercial quality honey), cockroaches (*Blaptica dubia*) and an artificial diet (**Bhatkar and Whitcomb, 1970**). For comparison, workers of one other *Camponotus* species (*Camponotus maculatus*), collected close to Kibale Forest, Uganda, in 2003 and housed under identical conditions as *Camponotus floridanus* were used. Additionally, six other formicine ant species, one *Lasius*, and five *Formica* species (*Lasius fuliginosus*, *Formica cinerea*, *Formica cunicularia*, *Formica fuscocinerea*, *Formica pratensis*, and *Formica rufibarbis*) were collected in Bayreuth, Germany in 2012 and 2018 and kept for approximately 2 weeks prior experimental use at 20°C, 70% humidity and a 14 hr/10 hr light/dark cycle. Except otherwise noted only the small worker caste ('minors') of *Camponotus* species was used.

## Acidity of the crop lumen and pH measurements

To determine whether formicine ants swallow their poison after feeding, we tracked changes in pH-levels of the crop lumen in *C. floridanus* ants over time. Before use in experimental settings, cohorts of ~100 ants were taken out of their natal colony (n = 6 colonies) into small plastic containers lined with Fluon and starved for 24–48 hr. Thereafter, ants were put singly into small petri dishes (Ø 55 mm) with damp filter paper covered bottom, given access to a droplet of 10% honey water (w/v) for 2 hr before removing the food source and measuring the pH of the crop lumen after another 2 hr (group 0+4 hr: n = 60 workers), after 24 hr (group 0+24 hr: n = 59 workers) or 48 hr (group 0+48 hr:

n = 52 workers). To assess the effect of renewed feeding, a separate group of *C. floridanus* ants were given access to 10% honey water 48 hr after the first feeding for 2 hr prior to measuring the pH of their crop lumen after another 2 hr (group 48h+4 hr: n = 60 workers). To measure the pH, ants were first cold anesthetized on ice, then their gaster was cut off with a fine dissection scissor directly behind the petiole and leaking crop content (1–3 µL) collected with a capillary (5 µL Disposable Micro Pipettes, Blaubrand intraMARK, Brand, Wertheim). The collected crop content was then emptied on a pH sensitive paper to assess the pH (Hartenstein, Unitest pH 1–11). This method of collecting crop content will invariably result in some mixing of crop lumen content with haemolymph. As the pH of the insect haemolymph ranges from only slightly acidic (pH $\geq 6.5$) to near-neutral or slightly alkaline (pH $\leq 8.2$) (*Harrison, 2001*; *Matthews, 2017*), this might have biased the results of our pH measurements to slightly higher pH values. As a reference point for food pH, we also measured the pH of 10% honey water on pH sensitive paper, which gave invariably pH = 5.

In addition, we measured the pH in the crop lumen and at four points in the lumen along the midgut (1st measurement directly behind proventriculus to 4th measurement one mm apical from insertion point of the Malpighian tubules) of *C. floridanus* workers that were fed 24 hr prior to measurements with 10% honey-water. For these measurements worker guts were dissected as a whole and pH was measured in the crop (n = 2 workers from two colonies) and along the midgut (all midgut points n = 10, except point four with n = 9 workers from four different colonies) with a needle-shaped microelectrode (UNISENSE pH-meter; microelectrode with needle tip of 20 µm diameter).

In formicine ants, oral uptake of the poison into the mouth is performed via acidopore grooming (*Tragust et al., 2013*). During this behavior ants bend their gaster forward between the legs and the head down to meet the acidopore, the opening of the poison gland, at the gaster tip (*Basibuyuk and Quicke, 1999*; *Farish, 1972*). In an additional experiment we therefore compared the crop lumen pH of *C. floridanus* workers from four different colonies that were either prevented to reach their acidopore (FA- ants) or could reach their acidopore freely (FA+ ants). To do this, we again allowed single ants access to 10% honey water for 2 hr after a starvation period, before cold anesthetizing them briefly on ice and immobilizing FA- ants (n = 22 workers) in a pipetting tip, while FA+ ants (n = 23 workers) remained un-manipulated. After 24 hr we measured the pH of the crop lumen as before.

To investigate whether swallowing of the acidic poison is widespread among formicine ants, the latter experiment was repeated for six additional formicine ant species (FA- ants: n = 10 workers except for *Formica pratensis* with n = 21; FA+ ants: n = 10 workers except for *Formica pratensis* with n = 20; all ants: n = 1 colony) in the same fashion as described before with the exception that apart from *Formica pratensis* the crop lumen was collected through the mouth by gently pressing the ants' gaster. Crop lumen of *Formica pratensis* ants was collected in the same fashion as crop lumen of *C. floridanus* ants.

To investigate whether the type of fluid and its nutritional value have an influence on the frequency of acidopore grooming in *C. floridanus*, the following experiment was performed. Cohorts of ~100 ants were taken out of their natal colony (n = 6 colonies) into small plastic containers and starved for 24–48 hr. Thereafter, ants were again put singly into small petri dishes (Ø 55 mm) and given access to either a 3 µL droplet of 10% honey water (w/v, n = 126 ants, treatment: honey-water fed), a 3 µL droplet of tap water (n = 128, water-fed) or to no fluid (n = 125, unfed). After acclimatization (unfed ants) or after swallowing of the fluid (honey-water and water-fed ants, both 1–2 min.), all ants were filmed for the next 30 min. (Logitech webcam c910). These videos were then analyzed for the frequency of acidopore grooming.

Finally, we measured the pH in the crop lumen of *C. floridanus* ants (n = 3 colonies) under satiated and starved conditions to estimate a baseline level of acidity in the crop. For this, ants taken out of satiated, twice per week fed colonies on the day of feeding were compared to ants that were maintained in cohorts of ~100 individuals for 3 d with access to 10% honey-water and then starved for 24 hr before measuring the pH in their crop (n = 10 major and 10 minor workers per colony and condition). The pH in the crop lumen was measured as described before by briefly cold anesthetizing ants an ice, collecting the crop content through the mouth by gently pressing the ants' gaster and then emptying it on a pH sensitive paper (Hartenstein, Unitest pH 1–11).

## Bacterial strains and culture

As model entomopathogenic bacterium *Serratia marcescens* DSM12481 (DSMZ Braunschweig, Germany) was used. This bacterium is pathogenic in a range of insects (*Grimont and Grimont, 2006*) and has been detected in formicine ants, that is *Anoplolepis gracilipes* (*Cooling et al., 2018*) and *Camponotus floridanus* (*Ratzka et al., 2011*). While often non-lethal within the digestive tract, *S. marcescens* can cross the insect gut wall (*Mirabito and Rosengaus, 2016*; *Nehme et al., 2007*) and is highly virulent upon entry into the hemocoel (*Flyg et al., 1980*), not least due to the production of bacterial toxins (*Hertle, 2005*). As a model bacterial gut-associate of ants *Asaia* sp. strain SF2.1 (*Favia et al., 2007*), was used. *Asaia* sp. belongs to the family Acetobacteraceae, members of which often thrive in sugar-rich environments (*Mamlouk and Gullo, 2013*), such as honeydew that ants like *C. floridanus* predominantly feed on. *Asaia* sp. is capable of cross-colonizing insects of phylogenetically distant genera and orders (*Crotti et al., 2009*; *Favia et al., 2007*) and can be a component of the gut-associated microbial community of formicine and other ants (*Chua et al., 2018*; *Kautz et al., 2013a*; *Kautz et al., 2013b*). In addition to *S. marcescens* and *Asaia* sp., *Escherichia coli* DSM6897 (DSMZ Braunschweig, Germany) was used as a model opportunistic bacterium that is not a gut-associate of insects. *E. coli* bacteria are a principal constituent of mammalian gut-associated microbial communities but are commonly also found in the environment (*Blount, 2015*).

Bacterial stocks of *S. marcescens*, *Asaia* sp., and *E. coli* were kept in 25% glycerol at −80°C until use. For use, bacteria were plated on agar plates (LB-medium: 10 g tryptone, 5 g yeast extract, 20 g agar in 1L MilliQ-water, and GLY-medium: 25 g gycerol, 10 g yeast extract, 20 g agar in 1L MilliQ-water with pH adjusted to 5.0, for *S. marcescens/E. coli* and *Asaia* sp. respectively), single colony forming units (CFUs) were picked after 24 hr (*S. marcescens/E. coli*) or 48 hr (*Asaia* sp.) of growth at 30°C and transferred to 5 ml liquid medium (LB-medium and GLY-medium minus agar for *S. marcescens/E. coli* and *Asaia* sp. respectively) for an overnight culture (24 hr) at 30°C. The overnight culture was then pelleted by centrifugation at 3000 g, the medium discarded and resolved in 10% (w/v) honey water to the respective working concentration for the experiments. The concentration of a typical overnight culture was determined for *S. marcescens* and *Asaia* sp. by plating part of the overnight culture on agar plates and counting CFUs after 24 hr or 48 hr of growth at 30°C, for *S. marcescens* and *Asaia* sp. respectively. This yielded a concentration of $1.865 * 10^9 \pm 5.63 * 10^7$ (mean ± sd) bacteria per mL for *S. marcescens* and $5.13 * 10^8 \pm 8.48 * 10^6$ (mean ± sd) bacteria for *Asaia* sp.

## Survival experiments

In a first survival experiment we tested whether the ability to perform acidopore grooming within the first 24 hr after ingestion of pathogen contaminated food provides a survival benefit for individual *C. floridanus* ants. Ants from eight colonies were starved for 24–48 hr before start of the experiment, as described before, and then workers put singly in small petri dishes were either given access to 5 µL of *S. marcescens* contaminated 10% honey water ($9.33 * 10^9$ bacteria/mL; *Serratia+* ants: n = 127) or uncontaminated 10% honey water (*Serratia-* ants: n = 135) for 2 min. Afterward, all ants were cold anaesthetized and approximately half of the *Serratia+* and the *Serratia-* ants (n = 65 and n = 69, respectively) immobilized in a pipetting tip, thus preventing acidopore grooming (FA- ants: n = 134) while the other half remained fully mobile (FA+ ants: n = 128). After 24 hr, FA- ants were freed from the pipetting tip to minimize stress. Mortality of the ants was monitored over 5 d (120 hr) every 12 hr providing no additional food, except the one time feeding of 5 µL contaminated or uncontaminated honey water at the start of the experiment. We chose to provide no additional food after the one time feeding at the beginning of the experiment, as an altered feeding behavior, that is, illness induced anorexia with known positive or negative effects on survival (*Hite et al., 2020*), might otherwise have influenced our results.

In an additional survival experiment, we investigated whether the ability to acidify the crop lumen has the potential to limit oral disease transmission during trophallactic food transfer. To this end, *C. floridanus* ants from seven colonies were again starved, divided randomly in two groups (donor and receiver ants, each n = 322) and their gaster marked with one of two colors (Edding 751). Additionally, to prevent uptake of the poison, the acidopore opening of all receiver ants (receiver FA-) and half of the donor ants (donor FA-) was sealed with superglue, while the other half of the donor ants were sham treated (donor FA+) with a droplet of superglue on their gaster (*Tragust et al., 2013*). We then paired these ants into two different donor-receiver ant pairs. Pairs with both donor and

receiver ants having their acidopore sealed (donor FA- | receiver FA-) and pairs with only receiver ants having their acidopore sealed (donor FA+ | receiver FA-). Six hours after pairing, donor ants from both pairs were isolated and given access to 5 µl of *S. marcescens* contaminated 10% honey water (1.865 * 10^9 bacteria/mL) for 12 hr. Thereafter donor ants were again paired with the respective receiver ants for 12 hr and all pairs filmed for the first 30 min (Logitech webcam c910). These videos were then analyzed for the duration of trophallaxis events donor-receiver ant pairs engaged in during the first bout of trophallactic food exchange. After this first feeding round, donor ants were fed in the same fashion, that is, isolation for 12 hr with access to *S. marcescens* contaminated 10% honey water, every 48 hr, while they were maintained with the respective receiver ants for the rest of the time. This experimental design ensured that receiver ants were fed only through the respective donor ants with pathogen contaminated food. Survival of both, donor and receiver ants, was monitored daily for a total of 12 d.

## Bacterial viability and growth assays

We tested the ability of *S. marcescens* and *Asaia* sp. to withstand acidic environments in vitro, as well as their ability and the ability of *E. coli* to pass from the crop to the midgut in vivo when ingested together with food. In ants, gut morphological structures, that is, the infrabuccal pocket, an invagination of the hypopharynx in the oral cavity (*Eisner and Happ, 1962*), and the proventriculus, a valve that mechanically restricts passage of fluids from the crop to the midgut (*Eisner and Wilson, 1952*), consecutively filter solid particles down to 2 µm (*Lanan et al., 2016*) which would allow *S. marcescens* (Ø: 0.5–0.8 µm, length: 0.9–2 µm, *Grimont and Grimont, 2006*), *Asaia* sp. (Ø: 0.4–1 µm, length: 0.8–2.5 µm, *Komagata et al., 2014*), and *E. coli* (length: 1 µm, width: 0.35 µm, *Blount, 2015*) to pass. For the in vitro tests we incubated a diluted bacterial overnight culture (10^5 and 10^4 CFU/ml for *S. marcescens* and *Asaia* sp., respectively) in 10% honey water (pH = 5) and in 10% honey water acidified with commercial formic acid to a pH of 4, 3, or 2 for 2 hr at room temperature (*S. marcescens*: n = 15 for all pH-levels, except pH = 4 with n = 13; *Asaia* sp.: n = 10). Then we plated 100 µl of the bacterial solutions on agar-medium (LB-medium and GLY-medium for *S. marcescens* and *Asaia* sp., respectively) and incubated them at 30°C for 24 hr (*S. marcescens*) or 48 hr (*Asaia* sp.) before counting the number of formed CFUs. For the in vivo tests *C. floridanus* ants from five (*Asaia* sp.), four (*E. coli*) or from six colonies (*S. marcescens*) were starved as before and then individually given access to 5 µL of bacteria contaminated 10% honey water (*Asaia* sp. and *E. coli*: 1 * 10^7 CFU/mL, *S. marcescens*: 1 * 10^6 CFU/mL) for 2 min. To assess the number of CFUs in the digestive tract, that is the crop and the midgut, ants were dissected either directly after feeding (0 hr; *S. marcescens*: n = 60 workers; *Asaia* sp. and *E. coli*: n = 15 each), or at 0.5 hr (*S. marcescens*: n = 60; *Asaia* sp. and *E. coli*: n = 15 each), 4 hr (*S. marcescens*: n = 60; *Asaia* sp. and *E. coli*: n = 15 each), 24 hr (*S. marcescens*: n = 53; *Asaia* sp. and *E. coli*: n = 15 each) or 48 hr (*S. marcescens*: n = 19; *Asaia* sp. and *E. coli*: n = 15 each) after feeding. For dissection, ants were cold anesthetized, the gaster opened and the whole gut detached. The crop and the midgut were then separated from the digestive tract, placed in a reaction tube, mechanically crushed with a sterile pestle and dissolved in 100 µL (*Asaia* sp. and *E. coli*) or 150 µL (*S. marcescens*) phosphate buffered saline (PBS-buffer: 8.74 g NaCl, 1.78 g Na$_2$HPO$_4$,2H$_2$O in 1L MilliQ-water adjusted to a pH of 6.5). The resulting solutions were then thoroughly mixed, 100 µl streaked on agar-medium (LB-medium and GLY-medium for *S. marcescens*/*E. coli* and *Asaia* sp., respectively) and incubated at 30°C for 24 hr (*S. marcescens* and *E. coli*) or 48 hr (*Asaia* sp.), before counting the number of formed CFUs. No other bacteria (e.g. resident microbes) were apparent in terms of a different CFU morphology on the agar plates which agrees with the very low number of cultivable resident bacteria present in the midgut of *C. floridanus* (Stoll and Gross, unpublished results). This methodology cannot completely exclude that resident *S. marcescens* or species of Acetobacteraceae might have biased our count data by adding a background level of CFUs at all timepoints or by adding random outlier CFUs at specific timepoints. Both, background level CFU numbers and random outlier CFUs should however not influence observed patterns over time. The timepoints of 0 hr, 0.5 hr, 4 hr, 24 hr, and 48 hr in in vivo bacterial growth assays were chosen according to literature describing passage of food from the crop to the midgut within 3–6 hr after food consumption in ants (*Cannon, 1998*; *Kloft, 1960b*; *Kloft, 1960a*; *Howard and Tschinkel, 1981*; *Markin, 1970*). They should thus be representative of two time points before food passage from the crop to the midgut (0.5 hr and 4 hr) and two time points after food passage from the crop to the midgut (24 hr and 48 hr) together with the reference timepoint (0 hr).

## Food passage experiment

To estimate food passage from the crop to the midgut and hindgut of *C. floridanus* after feeding we performed the following experiment. We again took a cohort of ~100 workers out of one natal colony of *C. floridanus*, starved them for 24 hr and then offered them 200 µL of a 1:1 honey-water mix with 50 mg of polymethylmethacrylate (PMMA, aka acrylic glass) particles (size ≤40 µm). Afterward, we dissected the digestive tract of three major and three minor workers at each of the timepoints 2 hr, 4 hr, 6 hr, 8 hr, 12 hr, 14 hr, 16 hr, 18 hr, 24 hr, and 48 hr after feeding and placed each under a microscope (Leica DM 2000 LED) to detect and count the number of particles via fluorescence in the crop, the midgut, and the hindgut.

## Statistical analyses

All statistical analyses were performed with the R statistical programming language (version 3.6.1, *R Development Core Team, 2019*). All (zero-inflated) General(ized) linear and mixed models and Cox mixed-effects models were compared to null (intercept only) or reduced models (for those with multiple predictors) using Likelihood Ratio (LR) tests to assess the significance of predictors. Pairwise comparisons between factor levels of a significant predictor were performed using pairwise post-hoc tests adjusting the family-wise error rate according to the method of Westfall (package 'multcomp', *Bretz et al., 2011*). We checked necessary model assumptions of (zero-inflated) General(ised) linear and mixed models using model diagnostic tests and plots implemented in the package 'DHARMa' (*Hartig, 2019*). Acidity of the crop lumen (log transformed pH to normalize data) and midgut lumen in *C. floridanus* was analyzed using linear mixed models (LMM, package"lme4', *Bates et al., 2015*) including time since feeding (four levels: 0+4 hr, 0+24 hr, 0+48 hr, 48h+4 hr; *Figure 1a*), ant manipulation (two levels: FA+ and FA-, that is ants with and without acidopore access; *Figure 1b*) or digestive tract part (four levels: crop, midgut position 1, midgut position 2, midgut position 3, midgut position 4; *Figure 1—figure supplement 1*) as predictors and natal colony as a random effect. Due to non-normality and heteroscedasticity, the acidity of the crop lumen in the seven formicine ant species other than *C. floridanus* (*Figure 1c*) was analysed using per species Wilcoxon rank-sum tests with ant manipulation (FA+ and FA-) as predictor. The frequency of acidopore grooming in *C. floridanus* upon feeding different types of fluids was analyzed using Generalized linear mixed models (GLMM, package"lme4', *Bates et al., 2015*) with negative binomial errors and type of fluid (three levels: unfed, water-fed, or 10% honey water fed) as predictor and natal colony as random effect (*Figure 1—figure supplement 2*).

Survival data were analysed with Cox mixed effects models (COXME, package 'coxme', *Therneau, 2019*). For the survival of individual ants (*Figure 3*), ant treatment (four levels: *Serratia-* | FA-, *Serratia-* | FA+, *Serratia+* | FA-, *Serratia+* | FA+) was added as a predictor and the three 'blocks' in which the experiment was run and the colony ants originated from, were included as two random intercept effects. For the survival of donor-receiver ant pairs (*Figure 4*), ant treatment (four levels: donor FA+, donor FA-, receiver FA+, receiver FA-) was included as a predictor and the three 'blocks' in which the experiment was run, the colony ants originated from, and petri dish in which donor and receiver ants were paired, were included as three random intercept effects. Survival of receiver ants was right censored if the corresponding donor ant died at the next feeding bout (right censoring of both donor and receiver ants in one pair upon death of one of the ants yielded statistically the same result: COXME, overall LR $\chi^2$ = 60.202, df = 3, p<0.001; post-hoc comparisons: receiver FA- versus donor FA-: p=0.388, all other comparisons: p<0.001). The duration of trophallaxis events (square-root transformed to normalize data) between donor-receiver ant pairs was analysed using a linear mixed model with ant pair type (two levels: donor FA+ | receiver FA- and donor FA- | receiver FA-) as predictor and the three 'blocks', in which the experiment was run and the colony ants originated from as random effect (*Figure 4—figure supplement 1*).

Bacterial growth in vitro was analysed separately for *S. marcescens* and *Asaia* sp. using Generalized linear models (GLM) with negative binomial errors and pH as predictor, excluding pH levels with zero bacterial growth due to complete data separation (*Figure 2—figure supplement 2* and *Figure 5—figure supplement 1*). Relative values shown in *Figure 2—figure supplement 2* and *Figure 5—figure supplement 1* were calculated by dividing each single number of formed CFUs at the different pH-values through the mean of formed CFUs at pH 5. Bacterial viability in vivo within the digestive tract of *C. floridanus* over time was analysed separately for the crop and midgut for *S.*

*marcescens* and *Asaia* sp. (*Figure 2* and *Figure 5*, respectively) and for *E. coli* (*Figure 2—figure supplement 2*). Zero-inflated generalized linear mixed models with negative binomial errors (package 'glmmTMB', *Brooks et al., 2017*) were used to model CFU number, with time after feeding as fixed predictor and ant colony as random effect, except for the *E. coli* model in the crop where colony was included as fixed factor as the model did not converge with colony as a random factor. Timepoints with zero bacterial growth were again excluded in the models. Relative CFU values shown in *Figure 2*, *Figure 5*, and *Figure 2—figure supplement 2* were calculated by dividing single CFU-values through the mean of CFU-values at timepoint 0 hr in the crop. Proportions and percentages of relative CFU change in the text are based on these relative CFU values.

## Acknowledgements

We would like to thank Robert Paxton for English grammar and style check of a pre-submission version of the manuscript, Franziska Vogel, Marvin Gilliar, and Martin Wolak for part of the data collection, Elena Crotti and Daniele Daffonchio for providing the *Asaia* strain and Martin Kaltenpoth for access to the pH microelectrode.

## Additional information

### Funding
No external funding was received for this work.

### Author contributions
Simon Tragust, Conceptualization, Resources, Data curation, Formal analysis, Supervision, Investigation, Visualization, Methodology, Writing - original draft, Project administration, Writing - review and editing; Claudia Herrmann, Jane Häfner, Ronja Braasch, Christina Tilgen, Maria Hoock, Margarita Artemis Milidakis, Investigation, Writing - review and editing; Roy Gross, Conceptualization, Writing - review and editing; Heike Feldhaar, Conceptualization, Supervision, Investigation, Methodology, Writing - review and editing

### Author ORCIDs
Simon Tragust (iD) https://orcid.org/0000-0002-3675-9583
Heike Feldhaar (iD) http://orcid.org/0000-0001-6797-5126

### Decision letter and Author response
Decision letter https://doi.org/10.7554/eLife.60287.sa1
Author response https://doi.org/10.7554/eLife.60287.sa2

## Additional files

### Supplementary files
• Source code 1. Script all analyses with R-source code. File containing all code required to reproduce the analyses and figures in R version 3.6.1.

• Transparent reporting form

### Data availability
The authors declare that all data supporting the findings of this study and all code required to reproduce the analyses and figures of this study are available within the article and its supplementary information and have been made publicly available at the DRYAD digital repository under the https://doi.org/10.5061/dryad.k0p2ngf4v.

The following dataset was generated:

| Author(s) | Year | Dataset title | Dataset URL | Database and Identifier |
|---|---|---|---|---|
| Tragust S | 2020 | Formicine ants swallow their highly acidic poison for gut microbial selection and control | http://dx.doi.org/10.5061/dryad.k0p2ngf4v | Dryad Digital Repository, 10.5061/dryad.k0p2ngf4v |

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
