## [Decision Letter]

**Acceptance summary:**

This paper identifies a new mechanism to regulate the microbiota in ants, linking crop acidification, bacterial differential survival and immunity.

**Decision letter after peer review:**

[Editors’ note: the authors submitted for reconsideration following the decision after peer review. What follows is the decision letter after the first round of review.]

Thank you for submitting your work entitled "Formicine ants swallow their highly acidic poison for gut microbial selection and control" for consideration by *eLife*. Your article has been reviewed by a Senior Editor, a Reviewing Editor, and three reviewers.

Our decision has been reached after consultation between the reviewers. Based on these discussions, and the individual reviews appended below, we have decided to reject your manuscript, but allow resubmission of a carefully-revised version with accompanying point-by-point response. This provides more than two months to prepare a manuscript that fully addresses the points raised by the reviewers. While this manuscript will be treated as a new submission, it will most likely be sent to the same reviewers and editors. Alternatively, if you think you will not be able to address all comments raised, we recommend you submit your manuscript for consideration elsewhere. Please note that resubmission to *eLife* does not guarantee eventual acceptance.

Briefly, all three reviewers liked the manuscript, but agreed that there were major concerns about:

1) potential side effects of the two different methods to block acidopore access;

2) baseline pH data in the crop and assessing the temporal extent of acidopore grooming beyond the 30 minutes to explain the continued decrease in pH;

3) lack of direct evidence for a decrease in disease transmission, as there are alternative explanations for the observed results;

4) the drop in Serratia survival in the crop above pH values that would be lethal in culture;

5) and the generally high mortality in the survival assays, even in pathogen-free ants.

It seems to us that most of these points can be addressed through more careful discussion and interpretation of the results that considers alternative explanations. Some points may, however, reflect serious flaws in the experimental design (especially points 1 and 5), but we can only assess this once we have received a revision and accompanying response.

Reviewer #1:

This study investigates a very interesting hypothesis that formicine ants may voluntarily swallow their own formic acid to acidify the content of their social stomach (the crop), which could act as a filter decreasing the risk of passage of live pathogenic microbes to the midgut, while allowing acidophilic symbiotic bacteria through. This mechanism would allow ants to solve the trade-off of allowing beneficial food-borne microbes in while simultaneously blocking harmful microbes and would play an analogous role as the stomach in higher vertebrates.

Overall, the study is of very high quality and builds a compelling set of evidence in favour of their working hypothesis, and is well worth publishing.

However, I have a few questions/concerns which I would like the authors to respond to, as I am not sure how they may affect the validity of the conclusions – and I feel they definitely should be addressed explicitly in the manuscript so the reader gets a full understanding of the implication of the study.

1) The main conclusion of the paper is that formicine ants actively swallow their poison gland secretions after feeding to acidify their crop. However, results from Figure 1A shows that the acidification of the crop continues much beyond 24 hours after feeding (pH significantly lower at 0h^+^48 hours than at 0h^+^24hours). To me this suggests that the acidification of the crop involves a continuous/constant mechanism, either via a physiological production of acid within the crop, or via a constant swallowing of poison gland secretion (which may be temporarily upregulated after feeding to compensate for the food dilution effect according to Figure 1—figure supplement 2). Unfortunately, measurements of acidopore grooming frequency were stopped 30 minutes after feeding so it is difficult to evaluate how long this upregulation lasts. However, I feel that the authors should rephrase their conclusions to avoid giving the impression that poison gland grooming occurs only after feeding (or that it is the only mechanism involved).

2) Key to demonstrating that crop acidification is a direct consequence of acidopore grooming are the experiments where the ants were prevented from grooming their acidopores (FA- ants).

2a) This is such an important part of the demonstration that I feel the methodology used to prevent acidopore grooming should appear in the main text (e.g. the Results and Discussion section), and not be 'hidden' within the Materials and methods section.

2b) When reading the Materials and methods section, I realised that in the first few experiments, the ants were prevented from grooming the acidopores by being immobilised inside a pipette tip, whereas control (FA+) ants were left to move freely. In my opinion this is the most problematic part of the study, as the two treatments differ by a lot more parameters than just acidopore grooming: compared to FA+, FA- ants cannot move, are under high levels of physiological stress, cannot interact with nestmates, cannot groom other parts of their bodies,…there are therefore a lot of alternative explanations for the differences between treatments. The authors need to acknowledge this weakness in their experimental design, and justify why they conclude that acidopore grooming is the one mechanism responsible for the observed differences

2c) In the last few experiments, FA- ants were obtained using a different protocole: application of superglue onto the acidopore. This method goes a long way towards addressing my concerns raised in 2b), and one wonders why the authors did not stick to the same procedure in all their experiments. However, when applying super-glue on the acidopore, there is a high risk of gluing the rectum and other glands shut at the same time, which could also have side-effects on survival of both donor and recipient ants. How can the authors be sure they only glued the acidopore shut? What precautions were taken to exclude ants onto which more glue was accidentally applied from the experiments? What is the consequences of this risk for the conclusions?

3) A key finding (shown in Figure 3—figure supplement 1) is that the growth of Serratia marcescens is inhibited by only 50% relative to pH5 under pH4 (reached at about 4 hours after feeding according to Figure 1A), and by almost 100% under pH3 (reached at about 24 hours after feeding according to Figure 1A). Similarly, Figure 3 shows that the amount of live S. marcescens in the crop has not decreased after 0.5hour after feeding but has decreased to almost 0 after 4 hours. For the crop to be effective as a filter, it seems indispensable that food remains inside the crop for a minimum of time (somewhere between 0.5hour and 4hours) before being passed to the midgut. This is a very important piece of information, yet it is only partially addressed at a late stage in the Materials and methods section(peak passage time of food from crop to midgut given in subsection “Bacterial growth assays”). I would like this to be moved to the main text, and more detail to be given (what is the minimum time within the crop?)

4) Can the authors discuss why the acidification did not extend to the midgut? What mechanism could prevent midgut acidification when food moves from the crop to the midgut?

5) Are there any acidophilic pathogenic bacteria known in ants? Or non-acidophilic symbiotic microbes found in the midgut? How does this fit with the main scenario?

6) Subsection “Statistical analyses”: which multiple-testing corrections were applied when using the Wilcoxon Rank Sum test over the 7 ant species?

7) In Figure 3A and 3C: I do not understand the legend for Figure 3 stating that what is displayed is the 'change in CFUs' relative to 0h in the crop. If that was the case, wouldn't all data-points for 0h in the crop be equal to 1 (if the 'change' is a ratio) or 0 (if the 'change' is a difference)? Please clarify.

8) Figure 3—figure supplement 1: same comment as above: if what is displayed is a 'change in CFUs relative to pH5', why aren't all points for pH5 equal to 1 or 0? Please clarify.

9) Some punctuation errors (there should be no comma after "both" in the Introduction, after "blocks" , after "colony", or after "petri dish" Subsection “Statistical analyses”). Some parts of the text should also be rewritten/simplified are they are difficult to follow (e.g. Results and Discussion section; "whether analogous to acidic…": grammatically incorrect in English; too wordy/hard to follow; hard to follow).

Reviewer #2:

This manuscript describes a novel mechanism of individual and social immune defense in ants, via the ingestion of acidic secretions from the abdominal poison gland. The authors carried out experiments demonstrating that (1) after feeding, the crop of Camponotus floridanus is acidified, but only if the ants have access to their abdomen; (2) the ant gut bacterium Asaia sp. survives acidic conditions in vitro and in vivo in the crop, whereas the pathogen Serratia does not, and neither does *E. coli*; (3) upon pathogen encounter, ants survive better when they have access to their poison glands, and nestmates also survive better when they interact with infected ants that have access to the poison gland than those that don't; and (4) crop acidification via ingestion of poison gland secretions appears to be widespread in formicine ants (demonstrated here in eight species across three genera). Although there are (less likely) alternative explanations for some of the results that could be discussed a bit more, the manuscript is generally very well-written and presents important novel findings that contribute to our understanding of individual-level and social immune defenses in ants. I have only one concern regarding the generally low survivorship of ants presented in Figure 2A, which in my view requires more explanation, but I anticipate the authors to be able to address this point. I commend the authors on a very interesting piece of work that presents exciting novel findings of broad interest.

Essential revisions:

1) The data presented in Figure 2A indicate that essentially all ants are dead after four days, regardless of whether they were exposed to Serratia or not, and the differences between treatment are in fact rather small. By contrast, the Serratia-exposed ants in Figure 2B lived much longer. What is the reason for the discrepancy in survival between the two experiments, and why do "healthy" ants die so quickly in Figure 2A?

Reviewer #3:

Summary:

This manuscript describes a series of experiments in carpenter ants that collectively show or suggest links between acidopore access, crop pH, and the survival of different bacteria in the gut.

That ants can use their own poison to adjust their crop pH in a way that selectively filters harmful bacteria is a fascinating idea. I enjoyed reading this manuscript, which is clearly written and data rich. The statistical analyses are appropriate and the authors provide all necessary raw data and clearly annotated R scripts to reproduce them.

The authors present convincing evidence for a link between acidopore access and crop pH, between acidopore access and survival following pathogen ingestion, and between pH and bacteria survival. They show more suggestive evidence for a role of acidopore grooming in limiting disease transmission and in 'filtering' harmful bacteria (while preserving presumed beneficial bacteria), with the causality of some links not entirely clear (see main comments below).

Essential revisions:

1) The results are interpreted as "prophylactic acidification" of the crop after feeding. Because no baseline level of acidity (i.e. before feeding) is provided, it is unclear whether acidopore grooming-induced changes in pH after feeding represent a transient acidification (i.e. from an otherwise higher 'normal' pH value), or a slow return to a baseline low pH (i.e. to maintain homeostasis) following a perturbation due to feeding. If the authors have such baseline (i.e. pre-feeding) data or can produce it easily, it would help interpret several results presented in the manuscript (e.g. it could help clarify point 3 below).

2) Figure 2B: The authors do not show direct evidence of a decrease in disease transmission, only an increase in survival when donor ants have acidopore access. In other words, while the differences in survival between receiver ants in the two treatments could plausibly be due to them receiving more bacteria from the donor ants, it could also be due to other effects (e.g. crop pH alone, donor overall health, etc.). To show a decrease in disease transmission would require showing differences in bacterial CFUs in the crop of receiver ants across treatments. I would therefore be very careful in interpreting these results in terms of disease transmission (e.g. Abstract, which currently reads "the ensuing creation of an acidic environment"… "limits disease transmission").

3) Figure 3: *S. marcescens* decreases to undetectable levels in the crop 4 hours post-feeding. The authors suggest that this decrease is due to low pH in the crop, itself presumably due to acidopore grooming (Results and Discussion section: "Consistent with this"). Based on the survival of *S. marcescens* at different pH values (Figure 3—figure supplement 1), this decrease would require crop pH 3 or lower. However, Figure 1A indicates that 4 hours post-feeding, crop pH is actually closer to 4. Can the authors address this seeming discrepancy? Currently, because the data shown in Figure 3 is not accompanied by gut pH data, it's difficult to clearly attribute the decline in *S. marcescens* to pH (rather than, say, immune responses).

[Editors’ note: further revisions were suggested prior to acceptance, as described below.]

Thank you for submitting your article "Formicine ants swallow their highly acidic poison for gut microbial selection and control" for consideration by *eLife*. Your article has been reviewed by three peer reviewers, and the evaluation has been overseen by a Reviewing Editor and Christian Rutz as the Senior Editor.

The reviewers have discussed their reviews with one another, and the Reviewing Editor has drafted this decision to help you prepare a revised submission.

Summary:

In this article, the authors provide evidence that formicine ants actively swallow their antimicrobial, highly acidic poison gland secretion to limit the establishment of pathogenic and opportunistic microbes ingested with food. This is an original mechanism to control the entry of pathogenic microbes.

Essential revisions:

1) All reviewers appreciated the care and effort that went into addressing the reviewers' earlier comments. However, we feel that the resulting doubling in length of the manuscript was not justified and actually made the article harder to read and the main message less clear. Some of the new material in the Discussion section almost amounts to mini reviews, which distract from -- and go beyond -- the scope of this article. For example, instead of a lengthy discussion of all the factors that can affect gut pH in insects, and of what is known of colony-wide patterns of trophallaxis in ants, it would be sufficient to briefly state that poison-swallowing is not the only way in which ants can adjust crop pH, and that the observed effects might have colony-wide effects, respectively. We would encourage the authors to trim back the article and be more synthetic when explaining caveats.

2) The reviewers still have one additional worry regarding the main conclusion of the article, namely, that the acidification of the crop acts like a filter by killing non-acidophilic pathogenic bacteria, but not acidophilic beneficial bacteria, before they are transferred to the midgut. The data presented provide indirect evidence that this is likely to be the case, but a key piece of the puzzle is missing to establish a causal relationship between acidification and filtering in vivo: namely, a demonstration that in the absence of acidification, a larger proportion of live bacteria is passed to the midgut (i.e., repeating the measurements shown in Figure 2 and Figure 5, but in immobilised ants or acidopore-blocked ants). Without that experiment, one cannot fully rule out the following alternative explanation: other immune mechanisms (but not acidification) are responsible for killing bacteria within the crop, and acidification is necessary for other biological functions, so that when ants are simultaneously faced with a bacterial challenge and a lack of acidification, the two deleterious effects combine to produce lower survival even in the absence of a direct effect of acidity on pathogen survival (this type of negative interaction between deleterious effects is often found in conservation studies where a combination of several threats leads to much faster extinction than any single threat would do). We are aware that an additional experiment may be difficult for the authors to perform at this stage, so we would like to offer them a choice. In case it is easy for them to do so, we would encourage them to repeat the measurements shown in Figure 2 and Figure 5 for acidopore-blocked or immobilised ants, as this would strengthen the article's conclusions as well as help shorten it, because some of the caveats currently detailed in the Discussion section would no longer need to be explored. Alternatively, we are still keen to publish the article, but we would then ask the authors to succinctly state in the Discussion section that their evidence on the effect of acidification is indirect and that they cannot rule out at present that other immune mechanisms are responsible for killing the pathogenic bacteria within the crop.

---

## [Author Response]

[Editors’ note: the authors resubmitted a revised version of the paper for consideration. What follows is the authors’ response to the first round of review.]

Briefly, all three reviewers liked the manuscript, but agreed that there were major concerns about:1) potential side effects of the two different methods to block acidopore access;2) baseline pH data in the crop and assessing the temporal extent of acidopore grooming beyond the 30 minutes to explain the continued decrease in pH;3) lack of direct evidence for a decrease in disease transmission, as there are alternative explanations for the observed results;4) the drop in Serratia survival in the crop above pH values that would be lethal in culture;5) and the generally high mortality in the survival assays, even in pathogen-free ants.It seems to us that most of these points can be addressed through more careful discussion and interpretation of the results that considers alternative explanations. Some points may, however, reflect serious flaws in the experimental design (especially points 1 and 5), but we can only assess this once we have received a revision and accompanying response.

In the revision, we have completely restructured the Results section and have added a distinct discussion where we carefully discuss and interpret our results considering alternative explanations. In addition, we have performed novel experiments that strengthen and extend our original conclusions.

Specifically, we now (1) clarify potential side effects and limitations of our methods to prevent acidopore grooming and to block acidopore access and explicitly state how we controlled for them in our experiments (answer to comment 2b and 2c of reviewer 1), (2) provide baseline pH data in the crop which indicates that swallowing of the poison and thus crop acidity is not a transient effect after perturbation of the crop pH through the ingestion of fluids but that ants rather aim to maintain an optimal, acidic pH in the crop (answer to comment 1 of reviewer 3), (3) carefully discuss that the improved survival of ants receiving pathogen contaminated food via trophallaxis from ants with the ability to swallow the poison can only provide indirect evidence for a decrease in disease transmission and that other effects might play a role (answer to comment 2 of reviewer 3), (4) explain that our choice to starve ants after a one time feeding of pathogen contaminated or non-contaminated food together with social isolation likely led to the overall high mortality in acidopore access prevented and non-prevented ants in the survival experiment shown in Figure 3 (answer to comment 1 of reviewer 2).

Reviewer #1:1) The main conclusion of the paper is that formicine ants actively swallow their poison gland secretions after feeding to acidify their crop. However, results from Figure 1A shows that the acidification of the crop continues much beyond 24 hours after feeding (pH significantly lower at 0h^+^48 hours than at 0h^+^24hours). To me this suggests that the acidification of the crop involves a continuous/constant mechanism, either via a physiological production of acid within the crop, or via a constant swallowing of poison gland secretion (which may be temporarily upregulated after feeding to compensate for the food dilution effect according to Figure 1—figure supplement 2). Unfortunately, measurements of acidopore grooming frequency were stopped 30 minutes after feeding so it is difficult to evaluate how long this upregulation lasts. However, I feel that the authors should rephrase their conclusions to avoid giving the impression that poison gland grooming occurs only after feeding (or that it is the only mechanism involved).

We agree that acidopore grooming and swallowing of the poison is a natural component of the behavioural repertoire of formicine ants that is constantly performed. This is indicated by the following:

1) Two previous studies investigating grooming behaviours over a range of hymenopteran families found that acidopore grooming is a specialized behaviour existing only in a subset of ant families, i.e. Formicinae, Dolichoderinae and Myrmicinae (Basibuyuk and Quicke, 1999, Farish, 1972). Behaviour in those studies was recorded for at least 15minutes. per species (Basibuyuk and Quicke, 1999: 15minutes. to 2hours; Farish, 1972: five recording session of 5minutes.) in field collected and lab raised animals (Basibuyuk and Quicke, 1999, not specified in Farish, 1972). Behavioural records in those studies were not performed under specific animal conditions, e.g. fed vs. unfed. Therefore, acidopore grooming occurs under a variety of conditions.

2) In line with this is the observation of the first author in a previous study (Tragust et al., 2013) that acidopore grooming occurs naturally in formicine ants nursing pupal brood and

3) The observation of the present study that acidopore grooming is not only performed after fluid ingestion but also occurs in unfed ants (Figure 1—figure supplement 2). The natural occurrence of acidopore grooming and swallowing of the acidic poison will inevitably lead to more acidic crop lumens over time as seen in Figure 1A and to acidic crop lumens in formicine ants as the natural state. According to the suggestion of reviewer 3 we have now added data on the baseline pH in crop lumens of formicine ants. This data revealed that the crop of *C. floridanus* workers is highly acidic irrespective of whether ants were taken directly out of a satiated colony or whether ant cohorts were satiated and then starved for 24hours before measurements (Figure 1—figure supplement 3). Therefore, we interpret the upregulated frequency of acidopore grooming after fluid ingestion (Figure 1—figure supplement 2) now as the ant’s pursuit to maintain an acidic, likely optimal baseline pH in their crop lumen after perturbation of the crop pH through the ingestion of fluids. We have included this interpretation in the result section and the Discussion section and avoid giving the impression that swallowing of the poison only occurs after feeding.

Regarding the reasoning why we think that acidopore grooming and swallowing of poison is the mechanism behind the acidity in crop of formicine ants we refer the reviewer to the answer given below under point 2b.

2) Key to demonstrating that crop acidification is a direct consequence of acidopore grooming are the experiments where the ants were prevented from grooming their acidopores (FA- ants).2a) This is such an important part of the demonstration that I feel the methodology used to prevent acidopore grooming should appear in the main text (e.g. the Results and Discussion section), and not be 'hidden' within the Materials and methods section.

We now mention the methodology to prevent acidopore grooming throughout the result section at appropriate places (Results section) together with a reference to the Materials and methods section for additional details.

2b) When reading the Materials and methods section, I realised that in the first few experiments, the ants were prevented from grooming the acidopores by being immobilised inside a pipette tip, whereas control (FA+) ants were left to move freely. In my opinion this is the most problematic part of the study, as the two treatments differ by a lot more parameters than just acidopore grooming: compared to FA+, FA- ants cannot move, are under high levels of physiological stress, cannot interact with nestmates, cannot groom other parts of their bodies,…there are therefore a lot of alternative explanations for the differences between treatments. The authors need to acknowledge this weakness in their experimental design, and justify why they conclude that acidopore grooming is the one mechanism responsible for the observed differences

Unfortunately, in experimental settings, especially with non-model organisms, it is often impossible to control for all relevant factors as their effect on the measure of interest is often unknown. We agree that the method of immobilising ants to prevent them from acidopore grooming differs as a treatment in more aspects than just the behaviour from ants that could move freely and that unknown factors that were not controlled for might influence the acidity of the crop of formicine ants. We now acknowledge this now throughout the discussion (see below) but none withstanding believe that our data provide a convincing case that swallowing of the poison is responsible for crop acidity in formicine ants for the reasons given below.

As rightly pointed out by the reviewer, immobilisation of ants likely results in an elevated level of stress and a lack of interaction with nestmates. The lack of interaction with nestmates should however not have played a role under the experimental settings in Figure 1, as all ants were kept singly in petri dishes irrespective of whether they were immobilised or not. Potentially elevated levels of stress are the reason why in the survival experiment (Figure 3), acidopore grooming prevented ants through immobilisation were freed again after 24hours, a timepoint after main food passage from the crop to the midgut (see added data Figure 2—figure supplement 1 and answer to point 3 below) to limit stress induced mortality. As immobilised ants receiving a noncontaminated food source in this experiment survived significantly better than immobilised ants receiving a pathogen contaminated food source, a survival that was not significantly different from ants that could move freely, we conclude that elevated levels of stress did likely not influence the results of this experiment (see the Discussion section). Instead we now point out in the discussion that starvation following the onetime feeding of contaminated and non-contaminated food together with social isolation likely explain the high mortality of ants observed in this experimental setup (see the Discussion section and answer given to point 1 of reviewer 2).

The inability to groom other parts of the ant’s body is precisely what we wanted to achieve with the methodology of immobilising the ants in pipetting tips for the following reasons:

1) As the reviewer pointed out under point 1 above, physiological mechanisms within the digestive tract or other internal sources (see the Discussion section) could potentially lead to acidic crops in formicine ants and immobilisation effectively eliminates a possible contribution of these internal sources while at the same time indicating that an external source is likely responsible for crop acidity. We have added this line of reasoning now in the Results section and the Discussion section.

2) Ants possess a diversity of exocrine glands. Some exocrine glands produce acidic secretions and could thus serve as external sources for crop acidity. We acknowledge this now in the Discussion section.

3) Most notable among these exocrine glands with respect to crop acidity, are the metapleural gland and the poison gland. Both produce acidic secretion in several ant species and are actively groomed with movements involving the mouth. Immobilisation effectively prevents the use of acidic substances from both glands and importantly for our comparative survey (Figure 1C), provides a comparable method to indicate acidic exocrine secretions as the most likely source for formicine ant crop acidity. In addition, acidic metapleural gland secretions could not have served as an external source for crop acidity in the focal ant species of our study, *Camponotus floridanus*, as ants of the genus *Camponotus* show, with few exceptions, an evolutionary loss of the metapleural gland (i.e. *C. floridanus* and *C. maculatus* used for our experiments do not possess a metapleural gland). Acidic metapleural gland secretions could however serve as sources for crop acidity in the *Formica* and *Lasius* ants tested in our comparative survey (Figure 1C). We now acknowledge this in the Discussion section together with other alternative explanations for the high variability in crop lumen acidity of immobilised and nonimmobilised ants in Figure 1C (see the Discussion section).

We agree that compared to immobilisation the application of superglue provides methodologically a more direct evidence for the acidic poison as the source for crop acidity in formicine ants. While immobilisation can only indicate external sources, most likely in the form of acidic exocrine secretions, the application of superglue on the acidopore directly hints at the acidic poison as external source. However, the poison is the most likely source with both manipulations in all formicine ants tested as a previous study of the first author (Tragust et al., 2013) provided evidence that acidopore grooming results in the uptake of the poison into the mouth of the formicine ant *Lasius neglectus* while metapleural gland grooming was never observed (see the Introduction and the Discussion section). In addition, as a method, both immobilisation as well as the application of superglue to the acidopore in survival experiments involving *Camponotus floridanus* ants (Figure 3 and Figure 4, respectively) yielded qualitatively the same results, i.e. a higher survival of unmanipulated ants, indicating that swallowing of acidic exocrine secretions/the acidic poison provides a fitness benefit to formicine ants after ingestion of pathogen contaminated food sources. In both survival experiments, the effect of ant manipulation was controlled for and is unlikely to explain survival differences (see above for survival shown in Figure 3 and see answer to point 2c below). The application of superglue and not immobilisation was used in the survival experiment involving donor and receiver ants (Figure 4) as it would have been impossible to perform the experiment with immobilisation.

2c) In the last few experiments, FA- ants were obtained using a different protocole: application of superglue onto the acidopore. This method goes a long way towards addressing my concerns raised in 2b), and one wonders why the authors did not stick to the same procedure in all their experiments. However, when applying super-glue on the acidopore, there is a high risk of gluing the rectum and other glands shut at the same time, which could also have side-effects on survival of both donor and recipient ants. How can the authors be sure they only glued the acidopore shut? What precautions were taken to exclude ants onto which more glue was accidentally applied from the experiments? What is the consequence of this risk for the conclusions?

We refer the reviewer to the reasoning outlined under point 2b for why we did not stick to the same procedure to prevent acidopore grooming in our experiments.

We completely agree that the application of superglue on the acidopore (Figure 4) or of superglue or nail varnish in diverse papers from other research groups (Tranter et al., 2014; Tranter and Hughes, 2015; Greystock and Hughes, 2011; Pull et al., 2018), will invariably not only block the efferent duct of the poison gland, but also the opening of the hindgut and the efferent ducts of other glands, namely the Dufour gland and the cloacal gland, as they all open into a cloacal chamber with the acidopore as the common opening (Hölldobler and Wilson, 1998; Wenseleers et al., 1998). Especially blockage of the hindgut and the ensuing inability to defecate is likely to have contributed in combination with the ingestion of *S. marcescens* contaminated honey water to mortality of ants shown in Figure 4. However, our experimental design controlled for this and all else being equal allows us to disentangle the contribution of ant manipulation, i.e. blockage *per se*, from the inability to access the poison upon contact with *S. marcescens* contaminated food. The contribution of ant manipulation *per se* to mortality in Figure 4 is given by the survival difference between unmanipulated donor ants (donor FA+ with direct access to the poison gland secretion; solid grey line in Figure 4) and manipulated receiver ants that obtained food through unmanipulated donor ants (receiver FA- ants with indirect access to the poison gland secretion through the donor ants; solid black line in Figure 4). The lower survival of receiver FA- ants compared to donor FA+ ants indicates that blockage of the acidopore with superglue results in elevated levels of mortality, a fact that we acknowledge in the Results section. The additional contribution of the inability to access the poison to blockage *per se* is given by the survival difference between manipulated receiver ants that obtained food through unmanipulated donor ants (receiver FA- ants with indirect access to the poison gland secretion through their donor ants; solid black line Figure 4) and manipulated receiver ants (receiver FA-; dashed black line Figure 4) that obtained food from manipulated donor ants (donor FA-; dashed grey line Figure 4), both with the inability to access the poison. The higher survival of receiver FA- ants that obtained food from donor FA+ ants compared to the survival of receiver FA- ants that obtained food from donor FA- ants indicates that access to the poison after feeding on pathogen contaminated food does not only improve survival of ants directly feeding on pathogen contaminated food but also of ants they share the contaminated food via trophallaxis. Hence, swallowing of the poison and the ensuing crop acidity in formicine ants has the potential to limit oral disease transmission during food distribution within the society (see the Results section and the Discussion section for a discussion thereof). Finally, the combined effect of ant manipulation *per se* and the inability to access poison gland substances is given by the mortality difference between unmanipulated donor ants (donor FA+; solid grey line in Figure 4) and ants in the manipulated donor-receiver ant pair (donor FA- and receiver FA-; dashed grey and black solid lines in Figure 2b, respectively), the latter two not differing in survival.

3) A key finding (shown in Figure 3—figure supplement 1) is that the growth of Serratia marcescens is inhibited by only 50% relative to pH5 under pH4 (reached at about 4 hours after feeding according to Figure 1A), and by almost 100% under pH3 (reached at about 24 hours after feeding according to Figure 1A). Similarly, Figure 3 shows that the amount of live *S. marcescens* in the crop has not decreased after 0.5hour after feeding, but has decreased to almost 0 after 4 hours. For the crop to be effective as a filter, it seems indispensable that food remains inside the crop for a minimum of time (somewhere between 0.5hour and 4hours) before being passed to the midgut. This is a very important piece of information, yet it is only partially addressed at a late stage in the Materials and methods section (peak passage time of food from crop to midgut given in subsection “Bacterial growth assays”). I would like this to be moved to the main text, and more detail to be given (what is the minimum time within the crop?)

Unfortunately, the literature on food passage from the crop to the midgut is limited and fragmentary in ants. Workers of the fire ant, *S. invicta*, will pass some of a 5% sucrose solution to the midgut within seconds of ingestion, but levels in the midgut are highest between six and 24hours after consumption (Howard and Tschinkel, 1981). In the wood ant, *F. polyctena*, only 20% of consumed radioactively labelled honey water is seen in the midgut 4.5 hours after feeding (Gößwald and Kloft, 1960a). In the argentine ant, *L. humile*, radioactively labelled food was mostly restricted to the crop for 3 to 6hours after feeding but had entered the midgut by 12hours (Markin, 1970). In the carpenter ant, *C. pennsylvanicus*, food labelled with sodium fluorescein revealed that food passage from the crop to the midgut was relatively stable 4-16hours after food ingestion and appeared to peak at 20hours. Moreover, in *C. pennsylvanicus*, fluorescent particles of a size of 0.5 -10 µm remained in the partly filled crop even at 20hours post-feeding (Cannon, 1998). We have now added a data set on the food passage from the crop to the midgut and hindgut of *Camponotus floridanus* over time (Material and methods section). To investigate food passage, we took a cohort of ~100 workers out of one natal colony of *C. floridanus*, starved them for 24hours and then offered them 200 µl of a 1:1 honey-water mix with 50mg of polymethylmethacrylate (PMMA, aka acrylic glass) particles (size ≤ 40 µm). Then, we dissected the digestive tract of three major and three minor workers at each of the timepoints 2hours, 4hours, 6hours, 8hours, 12hours, 14hours, 16hours, 18hours, 24hours and 48hours after feeding and placed each under a microscope (Leica DM 2000 LED) to detect and count the number of particles via fluorescence in the crop, the midgut and the hindgut. This data revealed that only few particles pass from the crop to the midgut until 2-4hours after feeding, while particle passage from the crop to the midgut steadily increased thereafter until 8h after feeding and then declined steadily (Results section). Thus, our data largely confirm literature reports on the timing of food passage in the gastrointestinal tract of ants (Discussion section).

Together, food passage data from the literature and our own experiments (new Figure 2—figure supplement 1), indicate that only a small amount of ingested food is passed from the crop to the midgut until 4h after feeding, while thereafter food is steadily passed from the crop to the midgut. Hence our decision to measure CFU numbers in Figure 2 at 0.5hour, 4hours, 24hours and 48hours in addition to the reference timepoint 0hour, is representative of two time points before main food passage from the crop to the midgut (0.5hour and 4hours) and two time points after main food passage from the crop to the midgut (24hours and 48hours). We realize that food passage information is especially important in the context of the ability of bacteria to withstand acidic conditions in the intestinal tract and have therefore added the new data as an additional figure supplement to Figure 2 (new Figure 2 —figure supplement 1).

We refer the reviewer to the answer given to point 3 of reviewer 3 why we think that the sensitivity of *S. marcescens* to withstand acidic conditions created with formic acid in vitro likely underestimates the antimicrobial effect of the formicine poison in vivo.

4) Can the authors discuss why the acidification did not extend to the midgut? What mechanism could prevent midgut acidification when food moves from the crop to the midgut?

As outlined under point 1 above, we interpret the baseline level of acidity in the crop (Figure 1—figure supplement 3) and the upregulated frequency of acidopore grooming after fluid ingestion (Figure 1—figure supplement 2) as the ants pursuit to maintain an optimal, acidic baseline pH in their crop after perturbation of the crop lumen pH through the ingestion of fluids. Although digestion is initiated in the crop, the midgut is the primary site of digestion in insects and the midgut epithelium plays a pivotal role in maintaining an optimal pH, as the gut pH is one of the most important regulators of digestive enzyme activity (Holtof et al., 2019, Terra and Ferreira, 1994). We think that this might be the reason why the midgut pH of *C. floridanus* shows only slightly acidic levels (pH 5) after highly acidic levels in the crop 24hours after feeding (see the Discussion section), a change of pH that might be achieved through physiological mechanisms (see the Discussion section for an outline how acidic conditions in the crop of insects might be achieved). In principle, a digestive compartment with a certain pH can be generated through physiological mechanisms involving a transport-loop of acid-base equivalents across epithelia (Onken and Moffett, 2017). Insects would thus regulate the pH of their crop or of other gut compartments through active uptake and excretion of acid–base equivalents across the gut epithelium (Matthews, 2017). Although insect gut compartments with extreme pH conditions have been reported in the literature, with few notable exceptions (Flower and Filshie, 1976, Miguel-Aliaga et al., 2018), the exact physiological mechanisms responsible for the creation of a gut lumen compartment with a certain pH are unknown in most insects (Harrison, 2001). We can thus only speculate, but physiological mechanisms involving active uptake and excretion of acid-base equivalents across the gut epithelium seem to us the most likely explanation for only slightly acidic conditions in the midgut of *C. floridanus*.

5) Are there any acidophilic pathogenic bacteria known in ants? Or non-acidophilic symbiotic microbes found in the midgut? How does this fit with the main scenario?

Some members of the Acetobacteraceae are pathogenic in humans and the fruit fly *Drosophila melanogaster* (Greenberg et al., 2006, Roh et al., 2008, Ryu et al., 2008) and most Acetobacteraceae produce metabolites that can potentially interfere with insect physiology and innate immunity (Chouaia et al., 2014). This may indicate that Acetobacteraceae found in formicine ants can act as pathogens. We have added this information in the Discussion section but are not aware that acidophilic pathogenic bacteria have been identified in ants.

Several groups of ants apparently harbour very few microbial associates, while others harbour a high density of microbial associates (Russel et al., 2017). Apart from specialised intracellular symbionts, these patterns have emerged only in recent years and detailed investigations on exact anatomical location, function and many other aspects are still unclear in most host-microbe associations involving ants. Many bacteria however produce short chain fatty acids and can thus acidify their environment (Ratzke et al., 2018, Ratzke and Gore, 2018). In particular, the Acetobacteraceae and various *Lactobacilli*, members of which are known as microbial associates of Hymenoptera (McFederick et al., 2013) release acetic acid as a waste product of their fermentative metabolism (Oude Elferink et al., 2001; Wolfe, 2005). Therefore, in ants and other animals that lack acidic poison gland secretions acidic derivatives produced by other gut microbial associates or environmental and defensive symbionts (Florez et al., 2015) might provide functionally similar roles to acidic poison gland secretions in formicine ants. Indications for this comes from studies in bees (Palmer-Young et al., 2018) and termites (Inagakie and Matsuura, 2018). We have added this line of reasoning in the Discussion section.

6) Subsection “Statistical analyses”: which multiple-testing corrections were applied when using the Wilcoxon Rank Sum test over the 7 ant species?

Crop acidity in the comparative study of the seven ant species was analysed with a Wilcoxon Rank Sum test for each species separately. As only ant treatment (two levels: FA- and FA+) was entered into the tests, no correction for multiple testing is needed.

7) In Figure 3A and 3C: I do not understand the legend for Figure 3 stating that what is displayed is the 'change in CFUs' relative to 0h in the crop. If that was the case, wouldn't all data-points for 0h in the crop be equal to 1 (if the 'change' is a ratio) or 0 (if the 'change' is a difference)? Please clarify.

Please see below our answer to the next comment.

8) Figure 3—figure supplement 1: same comment as above: if what is displayed is a 'change in CFUs relative to pH5', why aren't all points for pH5 equal to 1 or 0? Please clarify.

We apologize that we did not make this clear. Relative values shown in Figure 2, Figure 2—figure supplement 2, Figure 5, Figure 5—figure supplement 1 and Figure 5—figure supplement 2 were calculated by dividing all single CFU values through the mean of CFU-values of the reference level (Figure 2, Figure 5 and Figure 5—figure supplement 2: 0hour in the crop; Figure 2—figure supplement 2 and Figure 5—figure supplement 1: pH 5). This procedure was applied to all CFU-values and the obtained relative values are shown in the respective figures. The same procedure was also applied to values of the reference level, resulting in values bigger and smaller than one. We chose this calculation and representation, as it allowed us (1) to show variation in the obtained data also at the reference level, (2) to show patterns of change relative to the reference level and (3) to facilitate the comparison between the figures for the reader. Although we mentioned this calculation in the Materials and methods section of the previous version of the manuscript, we realize that without a proper explanation in the Results section and the figure legends, relative values appear puzzling. We therefore not only state the calculation in the Material and methods section but make it explicit in the Results section and the figure legends of Figure 2, Figure 2—figure supplement 2, Figure 5, Figure 5—figure supplement 1, and Figure 5—figure supplement 2.

9) Some punctuation errors (there should be no comma after "both" in the Introduction, after "blocks", after "colony", or after "petri dish" Subsection “Statistical analyses”). Some parts of the text should also be rewritten/simplified are they are difficult to follow (e.g. Results and Discussion section; "whether analogous to acidic…": grammatically incorrect in English; too wordy/hard to follow; hard to follow).

We thank the reviewer for pointing out these mistakes. We have amended them and tried to simplify sentences.

Reviewer #2:This manuscript describes a novel mechanism of individual and social immune defense in ants, via the ingestion of acidic secretions from the abdominal poison gland. The authors carried out experiments demonstrating that (1) after feeding, the crop of Camponotus floridanus is acidified, but only if the ants have access to their abdomen; (2) the ant gut bacterium Asaia sp. survives acidic conditions in vitro and in vivo in the crop, whereas the pathogen Serratia does not, and neither does *E. coli*; (3) upon pathogen encounter, ants survive better when they have access to their poison glands, and nestmates also survive better when they interact with infected ants that have access to the poison gland than those that don't; and (4) crop acidification via ingestion of poison gland secretions appears to be widespread in formicine ants (demonstrated here in eight species across three genera). Although there are (less likely) alternative explanations for some of the results that could be discussed a bit more, the manuscript is generally very well-written and presents important novel findings that contribute to our understanding of individual-level and social immune defenses in ants. I have only one major concern regarding the generally low survivorship of ants presented in Figure 2A, which in my view requires more explanation, but I anticipate the authors to be able to address this point. I commend the authors on a very interesting piece of work that presents exciting novel findings of broad interest.Essential revisions:1) The data presented in Figure 2A indicate that essentially all ants are dead after four days, regardless of whether they were exposed to Serratia or not, and the differences between treatment are in fact rather small. By contrast, the Serratia-exposed ants in Figure 2B lived much longer. What is the reason for the discrepancy in survival between the two experiments, and why do "healthy" ants die so quickly in Figure 2A?

We are sorry that differences in methodology between the two survival experiments were not clear in the previous version of the manuscript, though they are likely responsible for differences in survivorship between the two experiments. These differences include the (1) methodology to prevent access to the poison, (2) the feeding regime, (3) the social environment and (4) the dose of *S. marcescens* throughout the experiment outlined below in more details.

1) Ants in Figure 3 were prevented to access the poison for 24hours after exposure to *S.*

*marcescens* contaminated food through immobilisation in a pipetting tip. After 24hours they were released to minimize stress and potentially associated effects on mortality. Ants in Figure 4 were prevented access to the poison through blockage of the acidopore opening with superglue. As reviewer 1 rightly points out in her comment 2b and 2c, both methods might differ in more aspects than just the prevention poison access and might have other limitations (see our answer to these comments). The effect of these different ant manipulations on mortality rates was however controlled for with appropriate controls in the survival experiments. For the experiment in Figure 3 we found a nonsignificant difference between ants that could move freely and received a *S. marcescens* contaminated food source and ants that could move freely but received a noncontaminated food source. In contrast, we found a significant lower survival of immobilised ants fed a *S. marcescens* contaminated food source compared to immobilised ants that received a non-contaminated food source. As immobilised ants that received a non-contaminated food source did not differ in their survival from ants that could move freely, elevated levels of stress due to immobilisation are unlikely to have influence mortality in this experiment (see the Discussion section). For the experiment in Figure 4 we would like to direct the reviewer to the answer given to reviewer 1 to comment point 2c.

2) Ants in both survival experiments experienced an initial starvation period of 24-48hours before use under experimental condition. Thereafter, ants used in Figure 3 were only fed once with 5µl of a contaminated and non-contaminated food source at the beginning of the experiment while no additional food was given to them in the following days. *C. floridanus* ants thus experienced starvation under our experimental conditions which likely led to a high mortality in this experiment. In a similar experiment involving the formicine ant species *Formica exsecta*, oral exposure to *S. marcescens* contaminated food followed by starvation as well as starvation alone also led to a high mortality of ants with no additive effects of pathogen exposure combined with starvation (Stucki et al., 2019). Contrary to *C. floridanus* ants in Figure 3, ants in Figure 4 were fed every other day with 5µl of *S. marcescens* contaminated honey water (directly: donor ants, indirectly through trophallaxis: receiver ants) after the initial starvation period. Thus, ants in Figure 4 were continuously fed and likely experienced no or only a mild starvation. We have now mentioned these differences in experimental feeding regime explicitly throughout the Results section and have added in the Discussion section that starvation is one of the likely reasons for the high mortality of ants in Figure 3 irrespective of whether they received pathogen contaminated food or not (Discussion section).

3) In addition to starvation, ants in the experiment shown in Figure 3 were kept singly in petri dishes, while ants in Figure 4 were kept in pairs of donor and receiver ants and were kept singly only for 12hours every 48hours when donor ants were fed directly with *S. marcescens* contaminated food. Social isolation has been shown to increase mortality in ants (Koto et al., 2015) and social isolation has been shown to reduce an individual’s capacity to fight infections in other group living animals (Kohlmeier et al., 2016). Thus, social isolation experienced by *C. floridanus* ants under experimental conditions in Figure 3 is likely, in addition to starvation, to have led to a generally increased mortality of ants. We have added this reasoning in the Discussion section.

4) Ants in Figure 3 and Figure 4 were exposed to different concentrations of *S. marcescens* in contaminated food at the beginning of the experiment and in the case of ants in Figure 4 continuously every 48hours during the experiment (9.33 * 10^9^ bacteria/ml and 1.865 * 10^9^ bacteria/ml for Figure 3 and Figure 4, respectively). The difference in *S. marcescens* concentration between the two survival experiments is unfortunate and not the result of choice but rather a calculation error. None withstanding, ants in Figure 3 and Figure 4 also experienced a widely different exposure to *S. marcescens* contaminated food, i.e. a onetime exposure for ants in Figure 3 vs a continuous exposure every 48hours for ants in Figure 4. Together, these differences in methodology preclude a direct comparison of mortality between the two experiments. A direct comparison of ant mortality was however also never our intention, as the two experiments test different hypotheses. While we test in Figure 3. whether access to the poison can improve survival upon directly ingesting *S. marcescens* contaminated food, in Figure 4 we test whether access to the poison shows the potential to limit disease transmission during trophallactic food exchange. To acknowledge this fact, we have now restructured our result section and present the results of the survival experiments in two separate figures. We realize that the data presented in Figure 4 does not provide direct evidence for a decrease in disease transmission. We acknowledge this now in the Discussion section where we also present alternative explanations (see the Discussion section and answer to point 2 of reviewer 3 below).

Reviewer #3:Summary:This manuscript describes a series of experiments in carpenter ants that collectively show or suggest links between acidopore access, crop pH, and the survival of different bacteria in the gut.That ants can use their own poison to adjust their crop pH in a way that selectively filters harmful bacteria is a fascinating idea. I enjoyed reading this manuscript, which is clearly written and data rich. The statistical analyses are appropriate and the authors provide all necessary raw data and clearly annotated R scripts to reproduce them.The authors present convincing evidence for a link between acidopore access and crop pH, between acidopore access and survival following pathogen ingestion, and between pH and bacteria survival. They show more suggestive evidence for a role of acidopore grooming in limiting disease transmission and in 'filtering' harmful bacteria (while preserving presumed beneficial bacteria), with the causality of some links not entirely clear (see main comments below).Essential revisions:1) The results are interpreted as "prophylactic acidification" of the crop after feeding. Because no baseline level of acidity (i.e. before feeding) is provided, it is unclear whether acidopore grooming-induced changes in pH after feeding represent a transient acidification (i.e.from an otherwise higher 'normal' pH value), or a slow return to a baseline low pH (i.e. to maintain homeostasis) following a perturbation due to feeding. If the authors have such baseline (i.e. pre-feeding) data or can produce it easily, it would help interpret several results presented in the manuscript (e.g. it could help clarify point 3 below).

We thank the reviewer for pointing out that our data in the previous version of the manuscript did not directly address this issue. We have now added data on the baseline pH in the crop of *C. floridanus* ants. This data revealed that the crop lumen of workers is highly acidic irrespective of whether ants were directly taken out of a satiated colony or whether ant cohorts were satiated for three days and then starved for 24hours before measurements (Figure 1—figure supplement 3). As pointed out in the response to point 1 of reviewer 1 several lines of evidence indicate that acidopore grooming and swallowing of the poison is a natural component of the behavioural repertoire of formicine ants under diverse circumstances. Thus, under constant conditions the pH of the crop will become increasingly acidic over time as seen in Figure 1A. However, constant conditions as in Figure 1A, will rarely be experienced by ants for long, as the crop content will be regularly perturbed during feeding and food exchange. This is evidenced by a series of recent studies with *Camponotus sanctus* (Greenwald et al., 2015 and Greenwald et al., 2018) which found that foragers as well as non-foragers dynamically fill, empty and mix crop liquid content. Under natural conditions perturbation of the crop content and with this the crop pH will therefore be rather the norm than the exception. According to this and the new data on baseline pH levels in the crop, we interpret the upregulated frequency of acidopore grooming after fluid ingestion (Figure 1—figure supplement 2) as the ants pursuit to maintain an optimal, acidic baseline pH in their crop after perturbation of the crop pH through the ingestion of fluids. We have included this interpretation in the Results section and the Discussion section.

2) Figure 2B: The authors do not show direct evidence of a decrease in disease transmission, only an increase in survival when donor ants have acidopore access. In other words, while the differences in survival between receiver ants in the two treatments could plausibly be due to them receiving more bacteria from the donor ants, it could also be due to other effects (e.g. crop pH alone, donor overall health, etc.). To show a decrease in disease transmission would require showing differences in bacterial CFUs in the crop of receiver ants across treatments. I would therefore be very careful in interpreting these results in terms of disease transmission (e.g. Abstract, which currently reads "the ensuing creation of an acidic environment"… "limits disease transmission").

We agree with the reviewer that our current data does not provide direct evidence of a decrease in disease transmission but only an increase in survival when donor ants have acidopore access. Although this might translate into a decrease of disease transmission, we lack data supporting this. We have therefore rephrased the sentence to: “This indicates that swallowing of the poison after feeding on pathogen contaminated food does not only improve survival of formicine ants directly feeding on pathogen contaminated food but also of ants that share the contaminated food via trophallaxis. Hence, swallowing of the poison and the ensuing crop acidity have the potential to limit oral disease transmission during food distribution within a formicine ant society.” (Results section)

We now also acknowledge in the Discussion section that apart from crop acidity and fewer bacteria being passed on from donor to receiver ants, other effects might play a role. This section now reads in the Discussion section: “ In addition to improve survival of ants that directly ingested pathogen contaminated food, the ability of donor ants to access their poison also improved survival of receiver ants without access to their poison when receiver ants shared pathogen contaminated food during trophallactic food exchange. Although our experiments on the ability of *S. marcescens* to withstand acidic conditions in vivo and in vitro indicate that this is likely due to fewer viable bacteria that are passed on from donor to receiver ants during trophallactic food exchange, it remains to be established whether this is indeed the case or whether this might be due to obtaining trophallactic fluids with antimicrobial activity. Antimicrobial activity of formicine ant trophallactic fluids has been described in previous studies (Hamilton et al., 2011, LeBoeuf et al., 2016). These studies linked the antimicrobial activity of trophallactic fluids to the presence of proteins related to cathepsin D, a lysosomal aspartic protease that can exhibit antibacterial effector activity and the proteolytic production of antimicrobial peptides (Ning et al., 2018). Our results however suggest a major role of swallowing the acidic poison to the antimicrobial activity of trophallactic fluids in formicine ants. Future studies will need to disentangle the relative contributions of crop acidity, proteins related to cathepsin D and, as previously pointed out, other immune effectors that are released into the insect gut to the antimicrobial activity of formicine ant trophallactic fluids.”

Finally, we acknowledge in the Discussion section that at the colony level our data is limited in showing a decrease of disease transmission and have therefore added a section on trophallactic food exchange and its potential consequences to disease transmission with respect to crop lumen acidity in formicine ants. This section now reads: “ The sensitivity of the bacterial pathogen *S. marcescens* to acidic conditions and the fitness benefit bestowed to ants with direct and indirect access to the poison after feeding on or receiving of pathogen contaminated food, might also indicate that swallowing of the poison and the ensuing crop acidity can act as an important barrier to oral disease spread within formicine ant societies. In the formicine ant Formica polyctena, food passage from the crop to the midgut is dependent upon whether food is directly eaten or is transferred via trophallaxis with only 20% of the honey water consumed directly seen in the midgut 4.5 hours after feeding, while 77% of the honey water received during trophallaxis reaching the midgut 2.5 hours after feeding (Gösswald and Kloft, 1960a). Poison acidified crop lumens might therefore alleviate the cost of sharing pathogen contaminated food (Onchuru et al., 2018, Salem et al., 2015) and effectively counteract the generally increased risk of pathogen exposure and transmission associated with group-living (Alexander, 1974, Boomsma et al., 2005, Kappeler et al., 2015). On the other hand, trophallactic food exchange is a dynamic process that is dependent upon the functional role of the worker (Greenwald et al., 2015), food type (Buffin et al., 2011), and likely many other contexts. For example, it has repeatedly been reported that after a time of starvation food is distributed extremely quickly and efficiently via trophallaxis within an ant colony (Buffin et al., 2009, Gösswald and Kloft, 1960b, Markin, 1970, Sendova-Franks et al., 2010, Traniello, 1977, Wilson and Eisner, 1957). While this increases the threat of pathogen dissemination together with food, it has been suggested that dilution and mixing of food together with the existence of disposable ants specialised in food storage can mitigate the threat of pathogenic and noxious substances distributed together with food (Buffin et al., 2011, Sendova-Franks et al., 2010). Recently, it has been reported that ant social networks can plastically respond to the presence of a pathogen and that ants alter their contact network to contain the spread of a disease (Stroeymeyt et al., 2018). In our study, the first bout of trophallactic food exchange between donor and receiver ants was not affected by the manipulation of poison access. However, especially at later time points, potential changes in the amount of food transmitted cannot be excluded, as trophallaxis and feeding behaviour in general might depend on the infection status of ants engaging in trophallactic food exchange (Hite et al., 2020). In future studies, it will therefore be interesting to examine whether swallowing of the poison and the ensuing crop acidity truly acts as a barrier to oral disease spread within formicine ant societies, especially given the technical advances to track multiple individuals of a group simultaneously over time that have been made in recent years (Gernat et al., 2018, Greenwald et al., 2015, Imirzian et al., 2019, Stroeymeyt et al., 2018).”

3) Figure 3: *S. marcescens* decreases to undetectable levels in the crop 4 hours post-feeding. The authors suggest that this decrease is due to low pH in the crop, itself presumably due to acidopore grooming (Results and Discussion section: "Consistent with this"). Based on the survival of *S. marcescens* at different pH values (Figure 3—figure supplement 1), this decrease would require crop pH 3 or lower. However, Figure 1A indicates that 4 hours post-feeding, crop pH is actually closer to 4. Can the authors address this seeming discrepancy? Currently, because the data shown in Figure 3 is not accompanied by gut pH data, it's difficult to clearly attribute the decline in *S. marcescens* to pH (rather than, say, immune responses).

The apparent discrepancy between the in vitro and in vivo ability of *S. marcescens* to withstand acidic conditions (shown in Figure 2—figure supplement 2 and in Figure 2, respectively) is likely explained by substances other than formic acid contained in the poison gland or substances added by the Dufour gland. Although the main component of the formicine ant poison is formic acid (Schmidt, 1986), the poison gland also contains acetic acid, hexadecanol, hexadecyl formate, hexadecyl acetat (Lopez et al., 1993) and a small fraction of unidentified peptides (Osman and Brandner, 1961, Herrmann and Blum, 1968). Moreover, in formicine ants the poison is usually expelled together from the acidopore with contents of the Dufour gland (Reigner and Wilson, 1968, Schoeters and Billen, 1995, but see Billen, 1982), which serve as wetting agents for the poison (Löfquist, 1977, see also Kohl et al., 2001). In a previous study, investigating the poison expelled from the acidopore by the formicine ant Lasius neglectus, we could confirm this and additionally tested the antimicrobial activity of different components of the poison against an entomopathogenic fungus either singly or in combination in vitro (Tragust et al., 2013). We found that formic acid alone could explain 70% of the antimicrobial activity of the poison, but that the combination of formic acid with other components of the poison gland and the Dufour gland could explain 94%. Thus, the inability of *S. marcescens* to withstand acidic conditions lower than pH 4 in vitro likely underestimates the antimicrobial effect of the formicine poison gland secretion in vivo. We have added this information in the Discussion section.

In addition, we agree with the reviewer that we cannot exclude that additional immune effectors such as AMPs might contribute to the observed decline of *S. marcescens* in the crop and its inability to establish in the midgut and explicitly acknowledge this now in the discussion. The corresponding sections read (Discussion section): “In a previous study we found that formic acid alone could explain 70% of the antimicrobial activity of the poison against an entomopathogenic fungus, but that the combination of formic acid with other components of the poison gland and the Dufour gland could explain 94% (Tragust et al., 2013). In addition to the likely higher antimicrobial activity of the natural poison compared to the activity of formic acid alone, in vivo immune system effectors released into the gut lumen might contribute to the inability of *S. marcescens* to establish in the gastrointestinal tract of *C. floridanus*. Highly acidic stomachs in vertebrates and acidic midgut regions in the fruit fly *Drosophila melanogaster* serve together with immune system effectors microbial control and prevent infection by oral pathogens (Giannella et al., 1972, Howden and Hunt, 1987, Martinsen et al., 2005, Overend et al., 2016, Rakoff-Nahoum et al., 2004, Slack et al., 2009, Tennant et al., 2008, Watnick and Jugder, 2020). Future studies will need to investigate the contribution of immune system effectors released into the gut lumen to the rapid reduction of *S. marcescens* in the crop of *C. floridanus* and its inability to establish in the midgut.” and (Discussion section): “Antimicrobial activity of formicine ant trophallactic fluids has been described in previous studies (Hamilton et al., 2011, LeBoeuf et al., 2016). These studies linked the antimicrobial activity of trophallactic fluids to the presence of proteins related to cathepsin D, a lysosomal aspartic protease that can exhibit antibacterial effector activity and the proteolytic production of antimicrobial peptides (Ning et al., 2018). Our results however suggest a major role of swallowing the acidic poison to the antimicrobial activity of trophallactic fluids in formicine ants. Future studies will need to disentangle the relative contributions of crop acidity, proteins related to cathepsin D and, as previously pointed out, other immune effectors that are released into the insect gut to the antimicrobial activity of formicine ant trophallactic fluids.”

[Editors’ note: what follows is the authors’ response to the second round of review.]

Summary:In this article, the authors provide evidence that formicine ants actively swallow their antimicrobial, highly acidic poison gland secretion to limit the establishment of pathogenic and opportunistic microbes ingested with food. This is an original mechanism to control the entry of pathogenic microbes.Essential revisions:1) All reviewers appreciated the care and effort that went into addressing the reviewers' earlier comments. However, we feel that the resulting doubling in length of the manuscript was not justified and actually made the article harder to read and the main message less clear. Some of the new material in the Discussion section almost amounts to mini reviews, which distract from -- and go beyond -- the scope of this article. For example, instead of a lengthy discussion of all the factors that can affect gut pH in insects, and of what is known of colony-wide patterns of trophallaxis in ants, it would be sufficient to briefly state that poison-swallowing is not the only way in which ants can adjust crop pH, and that the observed effects might have colony-wide effects, respectively. We would encourage the authors to trim back the article and be more synthetic when explaining caveats.2) The reviewers still have one additional worry regarding the main conclusion of the article, namely, that the acidification of the crop acts like a filter by killing non-acidophilic pathogenic bacteria, but not acidophilic beneficial bacteria, before they are transferred to the midgut. The data presented provide indirect evidence that this is likely to be the case, but a key piece of the puzzle is missing to establish a causal relationship between acidification and filtering in vivo: namely, a demonstration that in the absence of acidification, a larger proportion of live bacteria is passed to the midgut (i.e., repeating the measurements shown in Figure 2 and Figure 5, but in immobilised ants or acidopore-blocked ants). Without that experiment, one cannot fully rule out the following alternative explanation: other immune mechanisms (but not acidification) are responsible for killing bacteria within the crop, and acidification is necessary for other biological functions, so that when ants are simultaneously faced with a bacterial challenge and a lack of acidification, the two deleterious effects combine to produce lower survival even in the absence of a direct effect of acidity on pathogen survival (this type of negative interaction between deleterious effects is often found in conservation studies where a combination of several threats leads to much faster extinction than any single threat would do). We are aware that an additional experiment may be difficult for the authors to perform at this stage, so we would like to offer them a choice. In case it is easy for them to do so, we would encourage them to repeat the measurements shown in Figure 2 and Figure 5 for acidopore-blocked or immobilised ants, as this would strengthen the article's conclusions as well as help shorten it, because some of the caveats currently detailed in the Discussion section would no longer need to be explored. Alternatively, we are still keen to publish the article, but we would then ask the authors to succinctly state in the Discussion section that their evidence on the effect of acidification is indirect and that they cannot rule out at present that other immune mechanisms are responsible for killing the pathogenic bacteria within the crop.

We agree that the proposed additional experiments would provide more direct evidence for a causal relationship between acidification and filtering. Unfortunately, at present and for the foreseeable future, we are unable to perform these experiments. Moreover, to exclude that acidification is necessary for other biological functions and to elucidate the contribution of other immune mechanisms to the inhibitory effect of poison acidified formicine ant crops, more experiments would be sensible in addition to the proposed experiments, which would delay resubmission incalculably.

We therefore opted, as suggested in the decision to our previous submission, to considerably shorten the manuscript, especially in the discussion and to succinctly state in the discussion that our evidence for a causal relationship is indirect and that other immune mechanisms might cause microbial filtering.